

# Cons-training tensor networks: Embedding and optimization over discrete linear constraints

Javier Lopez-Piqueres[1★] and Jing Chen[2†]

**1** Department of Physics, University of Massachusetts, Amherst, MA 01003, USA
**2** Zapata AI, 100 Federal Street, Boston, MA 02110, USA

★ jlopezpiquer@umass.edu , † yzcj105@gmail.com

## Abstract

In this study, we introduce a novel family of tensor networks, termed *constrained matrix product states* (MPS), designed to incorporate exactly arbitrary discrete linear constraints, including inequalities, into sparse block structures. These tensor networks are particularly tailored for modeling distributions with support strictly over the feasible space, offering benefits such as reducing the search space in optimization problems, alleviating overfitting, improving training efficiency, and decreasing model size. Central to our approach is the concept of a *quantum region*, an extension of quantum numbers traditionally used in $U(1)$ symmetric tensor networks, adapted to capture any linear constraint, including the unconstrained scenario. We further develop a novel canonical form for these new MPS, which allow for the merging and factorization of tensor blocks according to quantum region fusion rules and permit optimal truncation schemes. Utilizing this canonical form, we apply an unsupervised training strategy to optimize arbitrary objective functions subject to discrete linear constraints. Our method's efficacy is demonstrated by solving the quadratic knapsack problem, achieving superior performance compared to a leading nonlinear integer programming solver. Additionally, we analyze the complexity and scalability of our approach, demonstrating its potential in addressing complex constrained combinatorial optimization problems.

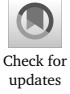

# 1  Introduction

Quantum physics has profoundly influenced the development of tensor network Ansätze and algorithms by leveraging entanglement area laws [1–5] and internal symmetries [6–9]. Building on the connection between linear equations and $U(1)$ symmetric tensor networks as detailed in Ref. [10], this study introduces a novel class of constrained tensor networks designed to embed both integer-valued equality and inequality linear constraints over binary variables. These constraints are central to combinatorial optimization problems characterized by the goal to optimize cost functions under specified linear conditions:

$$
\begin{aligned}
&\min C(\boldsymbol{x}) , \\
&\boldsymbol{\ell} \leq \mathbf{A}\boldsymbol{x} \leq \boldsymbol{u} , \\
&\boldsymbol{x} \in \{0,1\}^N , \\
&\boldsymbol{\ell}, \boldsymbol{u} \in \mathbb{Z}^M , \quad \mathbf{A} \in \mathbb{Z}^{M \times N} .
\end{aligned}
\tag{1}
$$

Our approach systematically handles these linear constraints through a constrained embedding step that builds a matrix product state (MPS) that parametrizes the entire feasible solution space exactly. This method leverages insights from constraint programming and the theory of $U(1)$ symmetric tensor networks, ensuring any optimization remains strictly within the feasible solution space. Unlike heuristic-based methods that often relax constraints through Lagrange multipliers or require post-processing to enforce constraints, our technique maintains hard constraints, resulting in an MPS with a block-sparse structure.

We utilize this constrained embedding step in conjunction with an optimization scheme that optimizes a model distribution constructed from the MPS in an unsupervised fashion through multiple iterations. At each iteration, training samples are drawn from a Boltzmann distribution based on the output model samples from previous iterations, with the probability for each training sample weighted by its cost function value and a temperature prefactor that is annealed over many cycles. This annealing process progressively concentrates the sampling probability towards the optimal solution, achieving progressively lower cost values from model samples throughout these iterations.

The work is divided into the following sections. In Sec. 2 we give an introduction to $U(1)$ symmetric tensor networks, and how they can be used to embed arbitrary equality constraints. Here we also introduce the notation used in the rest of the work. In Sec. 3 we extend the formalism of $U(1)$ symmetric matrix product states to encode arbitrary linear constraints, including inequalities. In Sec. 4 we analyze the efficiency of the proposed tensor network. A key concept here is the charge complexity of the tensor network ansatz. In Sec. 5 we introduce a novel canonical form for the proposed tensor network. In Sec. 6 we apply the constrained embedding step and the canonical form to optimize arbitrary cost functions subject to linear constraints via an unsupervised training strategy. We present results of this algorithm in Sec. 7. We give conclusions and outlook for future work in Sec. 8. An overview of the main results presented in this work is shown in Fig. 1. The code used to reproduce all numerical results is available in Ref. [11].

## 2 Symmetric tensor networks and quantum numbers

In this section we give a brief introduction to $U(1)$ symmetric matrix product states (MPS) and explain how do they relate to discrete linear equations of the form $\mathbf{A}x = b$, where $\mathbf{A} \in \mathbb{Z}^{M \times N}$, $b \in \mathbb{Z}^M$. We present two ways of viewing this encoding of linear equations into an MPS: a conceptual one based on finite state machines, and a more procedural one based on backtracking, a method widely used in constraint programming. For more details on the connection between linear equations and tensor networks we refer to Ref. [10], as well as the original work by Singh *et al.* on the general theory of $U(1)$ symmetric tensor networks [6].

### 2.1 Introduction to $U(1)$ symmetric matrix product states

While a common perspective is to view tensors appearing in MPS as mere multidimensional arrays, for discussing block-sparse tensors that arise from equalities like the one above, it is helpful to take the perspective of representation theory. In this regard, we view each tensor as a linear map between an *input* $\mathbb{V}^{\text{in}}$ and *output* vector space $\mathbb{V}^{\text{out}}$. This can be visualized by assigning arrows to each index of each tensor, as in Fig. 2.

A well-known result from representation theory dictates that, if such linear transformation is symmetric w.r.t. to a generic, compact group $\mathcal{G}$, it must map between states of fixed irrep. Each irrep is labeled by a well-defined *charge* or *quantum number* (QN), $n_i \in \mathbb{Z}$. If tensor $T$ is symmetric w.r.t. $U(1)$, then

$$U_\alpha^\dagger \otimes U_a^\dagger \otimes U_\beta T = e^{-in\phi} T, \tag{2}$$

with $U_x = e^{-i\hat{n}_x \phi}$, with $\phi \in [0, 2\pi)$. Here, $\hat{n}_x$ is the eigenoperator of the irrep labeled by charge $n_x \in \mathbb{Z}$: $\hat{n}_x |n_x\rangle = n_x |n_x\rangle$ and $|n_x\rangle \in \mathbb{V}_{n_x}$. We are taking the transpose conjugate of the representation $U_x$ when acting on input states. The integer $n$ denotes the total charge of the tensor and satisfies $n_\beta - n_a - n_\alpha = n$. Because of this property, the total charge is also known as the *flux*: *i.e.* the sum of outgoing minus incoming charges must match the total charge. The upshot is that $U(1)$ symmetry induces a block-sparse structure on $T$, with blocks labeled by charges fulfilling the charge conservation condition. Moreover, each block can be of any dimension, since each irrep $n_i$ can be of dimension $d_{n_i}$.

This formalism can be applied to higher dimensional tensors, such as MPS. In particular, an MPS is $U(1)$ symmetric if each of its tensors transforms as in Fig. (2). When an MPS is *charged*, its global charge or flux is usually carried by one of the tensors alone, denoted as the *flux tensor*. Moreover, when the MPS is in canonical form [12], it is also customary to assign such tensor as the canonical tensor.

**(a) Constrained Optimization**          **(b) Constrained Embedding**

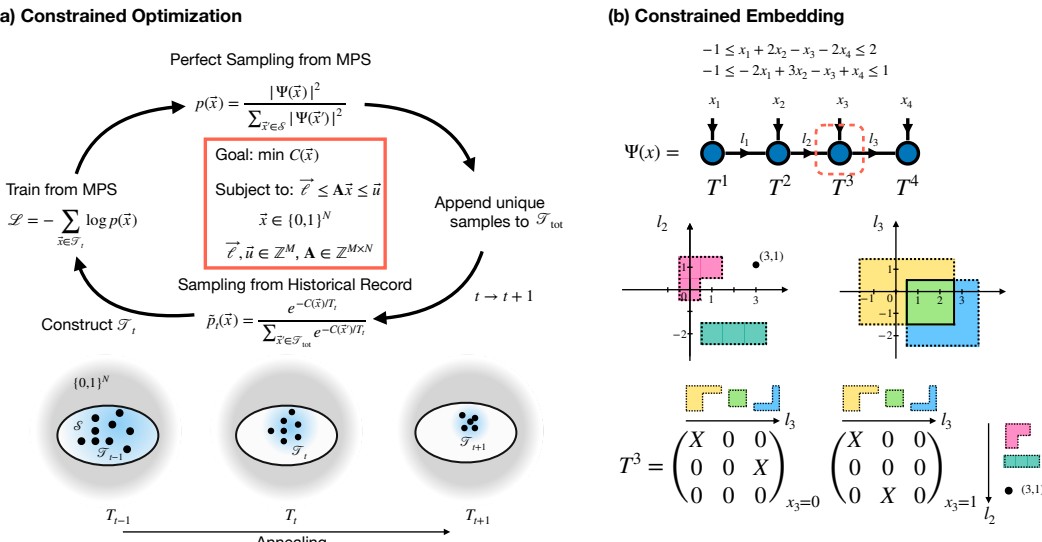

Figure 1: **Overview of the main contributions of this work.** *(a) Constrained Optimization*: We employ a model distribution constrained over the feasible solution space, $\mathcal{S} = \{x \in \{0,1\}^N : \ell \leq \mathbf{A}x \leq u\}$, using a new family of matrix product states (MPS) that we refer to as constrained MPS. This model is optimized through an unsupervised training strategy where training samples are drawn from a Boltzmann distribution based on historical model samples, with the temperature gradually annealed after each cycle. Initially, the sampling probability is relatively diffuse within the feasible space (ellipse in figure) at high $T$, but with annealing, the sampling probability increasingly concentrates towards the optimal solution. Our work introduces a novel canonical form for constrained MPS, enabling efficient training and perfect sampling on the MPS. The output samples from the MPS are then added to a dictionary containing samples from all previous iterations. This series of cycles is repeated until a stopping criterion is satisfied (*e.g.* time budget). *(b) Constrained Embedding*: Given $M$ linear constraints, the construction of constrained MPS requires link indices being parameterized by $m \leq M$ dimensional regions, *QRegions*. Shown an example with two inequalities. QRegions are found via an algorithm inspired by the theory of $U(1)$ symmetric tensor networks and the backtracking algorithm commonly used in constraint programming. MPS tensors are *quasi* block-sparse with blocks arranged according to a fusion rule for QRegions (see main text).

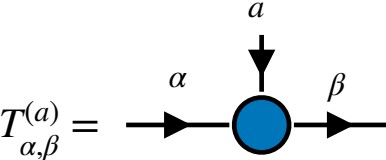

Figure 2: **MPS tensor as a linear map.** Rank-3 tensor with two incoming vector spaces labeled by $a$, $\alpha$, and one outgoing labeled by $\beta$. In this work we denote each physical (vertical) index as a superscript in parenthesis.

(a) $x_1$ $x_2$ $x_3$ (b)

$$T^{2(x_2)} = \begin{pmatrix} X & 0 & 0 \\ 0 & X & 0 \end{pmatrix}_{x_2=0} \quad \begin{matrix} l_1 \\ \end{matrix} \begin{matrix} 0 \\ 1 \end{matrix}$$

$$\begin{pmatrix} 0 & X & 0 \\ 0 & 0 & X \end{pmatrix}_{x_2=1} \quad \begin{matrix} l_1 \\ \end{matrix} \begin{matrix} 0 \\ 1 \end{matrix}$$

$l_1 = x_1$
$l_2 = l_1 + x_2$
$l_3 = l_2 + x_3$
$\quad = x_1 + x_2 + x_3$

Figure 3: **$U(1)$ symmetric MPS.** *(a)* Charge conservation in $U(1)$ symmetric MPS. Each *physical* (vertical) index carries a charge $x_i \in \{0, 1\}$. The charge conservation of the dashed green and orange regions can be guaranteed by local tensor conservation. The sum of all local charges must match the total charge or flux of the MPS, shown in dashed line. *(b)* The block structure of tensor at the second site, with each $X$ an unspecified block.

When tensors in an MPS collectively satisfy the charge conservation condition, contracting these tensors together maintains global conservation. As demonstrated in Fig. 3(a), the input charge equals the output charge across each local region, ensuring that the overall system respects $U(1)$ charge conservation.

All in all, by leveraging $U(1)$ symmetric tensor networks, we guarantee computational savings, since each tensor will be block-sparse (Fig. 3(b)).

## 2.2 Encoding arbitrary equality constraints in $U(1)$ symmetric matrix product states

Following our introduction to $U(1)$ symmetric matrix product states, we now explore their application in encoding the feasible solution space of linear equations. We transition from using specific charge indices $n_i$ to a more general labeling $l_i$ for link charges. Consider an equality constraint of fixed cardinality

$$x_1 + x_2 + x_3 = 2, \quad x_i \in \{0, 1\}. \tag{3}$$

This constraint mirrors constraints in quantum many-body physics, such as particle number conservation in bosonic systems or magnetization in spin chains, where the total number or magnetization remains fixed.

To construct the MPS corresponding to this equation we set the flux of the MPS of value 2 at the last site and set out to determine the charges on all links, $l_i$, consistent with charge conservation. This process can be conceptually likened to a finite state machine (FSM), where each link charge $l_i$ represents the state at step $i$, and each $x_i$ serves as the input at that step, see Fig. 4.

The state transitions within this FSM are dictated by the conservation of charge:

- If $x_i = 0$, the state remains unchanged ($l_{i+1} = l_i$).

- If $x_i = 1$, the state advances ($l_{i+1} = l_i + 1$).

To ensure the FSM concludes at $l_3 = 2$, various paths (colored in green, orange, and purple in Fig. 4(c)) represent all valid transitions from $l_0 = 0$ to $l_3 = 2$, corresponding to bit strings 110, 101, and 011.

We present now an effective method to derive link charges. This involves:

- *Determining Bounds:* Establish cumulative lower and upper bounds on the charge at each link, as illustrated in Fig. 4(c) in red and blue.



- *Recursive Solution:* Begin with the boundary condition $l_3 = 2$ and recursively solve *backwards* $l_i + x_{i+1} = l_{i+1}$. For instance, from $l_3 = 2$ and potential $x_3$ values, derive possible $l_2$ values within bounds.

- *Consistency Check:* In general, another *forward* pass from the left end is needed to remove all quantum numbers that are not consistent with the boundary condition $x_1 = l_1$.

This analysis can be carried out for any type of linear equation, and in fact, for any number of them [10]. This procedure bears resemblance to backtracking in the context of constraint programming, where solutions are explored systematically, reverting if constraints are violated. Crucially, however, our approach constructs the solution space indirectly by determining the relevant charges rather than directly computing all possible solutions. In the $U(1)$ symmetric MPS framework, this method involves defining link charges $l_i$ that are consistent with global charge conservation. Since the number of *local* unique charge configurations $\{l_i\}$ is generally fewer than the total number of potential solutions $\{x | \mathbf{A}x = b\}$, our method benefits from a more efficient encoding of the solution space, compared to traditional backtracking, which must consider each possible solution independently.

# 3 Constrained tensor networks and quantum regions

In this section, we generalize the formalism of $U(1)$ symmetric MPS to inequality constraints by introducing the concept of *QRegion*, short for *Quantum Region*. This consists of a group of QNs in the $M$-dimensional hyperplane (for $M$ constraints) effectively describing each link charge in the MPS. We start with a one inequality example of cardinality type and explain the motivation and benefits of grouping QNs into QRegions. As in the equality case, we provide two perspectives to the QRegion finding process: One conceptual based on FSMs and a procedural one based on backtracking. We then show how can we generalize these results in the presence of multiple inequalities, and present one of the main results of this work, captured by Algorithms 1 and 2 for constructing MPS from arbitrary linear constraints. We dub this new family of MPS *constrained MPS*.

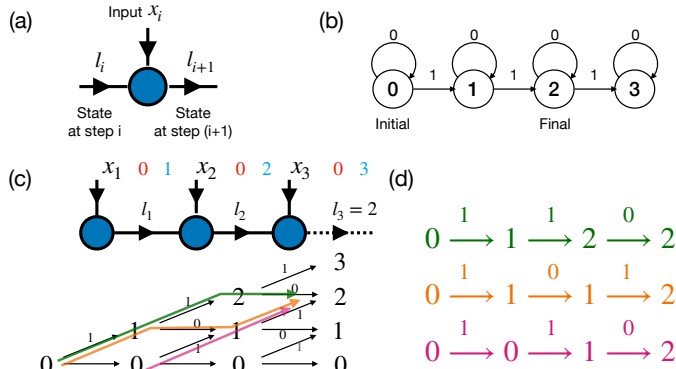

Figure 4: **Charge conservation as a finite state machine (FSM).** *(a)* The charge on each link is associated with a state. *(b)* We can view such states and their transitions as being part of a FSM. Shown here a FSM with final state 2. *(c)* *(Top)* MPS with flux 2 at the last link. Red/blue numbers on top of each link denote cumulative lower/upper bounds as we go left-to-right. *(Bottom)* If the finite state is constrained to 2, there are in total three transition paths colored by green, orange and purple. *(d)* Each of these paths corresponds to one feasible solution bitstring to $x_1 + x_2 + x_3 = 2$.

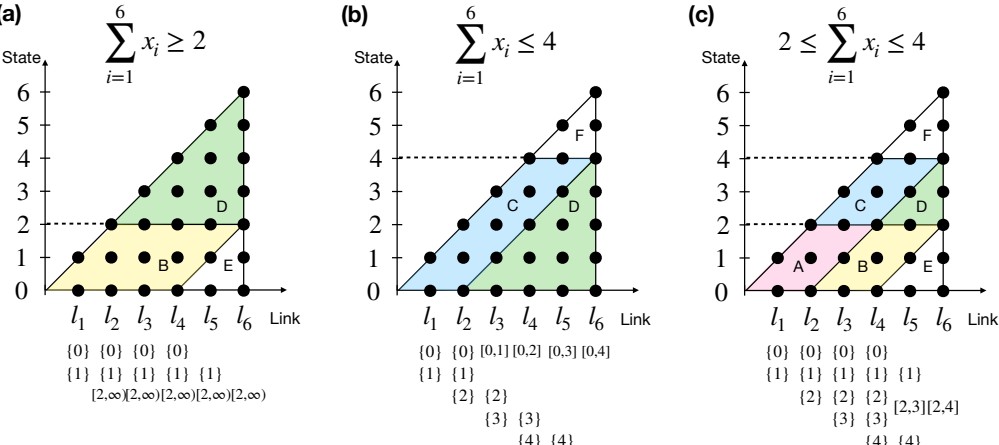

Figure 5: **Possible states of FSM of each link index with different constraints.** *(a)* Inequality $\sum_{i=1}^{6} x_i \geq 2$. *(b)* Inequality $\sum_{i=1}^{6} x_i \leq 4$. *(c)* Inequality $2 \leq \sum_{i=1}^{6} x_i \leq 4$. In each of these panels we show the set of allowed QNs at each link, and group them into a segment if they fall within the region shaded in green. All regions fall within the cumulative lower and upper bounds, indicated by the right triangle. More details on the meaning of each colored region is provided in the main text.

## 3.1 One inequality

Having demonstrated the computational advantages of utilizing $U(1)$ symmetric TNs for managing equality constraints, we now shift our focus to inequality constraints. A prevalent technique, widely used in the context of quadratic unconstrained binary optimization (QUBO) problems, involves transforming each inequality into an equality by incorporating slack binary variables [13]. Using this method, the number of extra binary variables grows with the number of inequalities, potentially increasing the complexity of the tensor network ansatz.

We propose here a more direct approach that encodes inequalities directly within the network's architecture, thereby reducing both the number of binary variables and potentially the bond dimension of the tensor network. Our approach is inspired by $U(1)$ symmetric TNs. Consider the inequality

$$2 \leq x_1 + x_2 + \cdots + x_6 \leq 4. \tag{4}$$

Rather than treating potential $U(1)$ fluxes separately for each case within the bounds, we encode them as a single interval $[2,4]$.

**Finite state machine approach:** We utilize a FSM strategy to dynamically manage the state transitions based on the sum $x_1 + x_2 + \cdots + x_i$. This approach, illustrated in Figure 5, allows for an *early-stop* strategy. This strategy halts computations when the constraints are definitively satisfied or violated, thereby avoiding unnecessary calculations. In our analysis, the cumulative lower and upper bounds correspond to setting all $x_i = 0$ and $x_i = 1$, respectively.

In view of a FSM, $l_0 = 0$ is the initial state, and the transition between states satisfying $l_i = l_{i-1} + x_i$. Since each $x_i \in \{0,1\}$, we have $0 \leq l_i \leq i$, which is in the triangular region with all values shown by points in Fig. 5. The boundary of which is provided by the cumulative upper and lower bound in Eq. 4 as we go from the first bit $x_1$ to the last $x_6$. In Fig. 5 we show three kinds of inequalities: Fig. 5(a) with only lower bound, 5(b) with only upper bound, and 5(c) with both lower and upper bound.

In (a) and following the early-stop policy, once the $l_i$ reaches 2, the constraint is unconditionally fulfilled. So we group any quantum numbers larger than 2 into one object, the

segment $[2, \infty)$. The corresponding region in (a) is labelled by the green region and symbol D. In contrast, the points in white E region would violate the constraint whatever the remaining $x_i, x_{i+1}, \cdots, x_6$ values are. The yellow region in B consist of charges that may or may not end up fulfilling the global constraint as this will depend on the remaining bits, so we cannot resolve them and we need to keep them all separate.

Similarly, in (b) we plot the case when only the upper bound constraint is considered. As long as the charge $l_i \leq i - 2$, we're guaranteed to fulfill the constraints. If by any chance, it enters the white F region, then it would already violate the constraint. Analogous to the yellow B region in (a), blue C region consists of charges that may or may not fulfill the global constraint so we cannot a priori group them.

If we combine these two cases together, and consider both constraints at the same time, we get panel (c). In this case we get the same colored regions as in panels (a) and (b), as well as a new region A in red, which is the intersection of regions B and C and is composed by QNs that cannot automatically be grouped. The goal is for the finite state $l_6$ to stay in the interval $[2, 4]$. The color code means the same as (a) and (b). The B (C) region means the upper (lower) bound is guaranteed to be fulfilled whatever the following $x_i$. However, satisfying the lower (upper) bound depends on the remaining $x_i$.

**Backtracking approach:**   Let us now show how these different regions are defined and segmented. Starting from the right-most link index $l_6$ and going to the first link index $l_1$, we call this step *backward decomposition* sweep, since the original segment $l_6 = [2, 4]$ is broken into smaller segments as we go left. To see this, we first resolve the relationship $l_5 \subseteq l_6 - x_6$. This computation yields two scenarios based on the values of $x_6$: For $x_6 = 0$, $l_5^{(x_6=0)} = [2, 4]$; for $x_6 = 1$, $l_5^{(x_6=1)} = [1, 3]$. So $[2, 4]$ and $[1, 3]$ have some overlaps, while also differences. So it's reasonable to split the interval into a disjoint union of non-overlapping and overlapping segments. This guarantees computational savings by grouping overlapping terms. In this case, we get 1, $[2, 3]$, 4. These belong to regions B (yellow), D (green) and C (blue) regions, respectively. These three items form $l_5$, *i.e.* $l_5 = \{1, [2, 3], 4\}$. Hence the original segment $[2, 4]$ has been broken into smaller pieces. In an analogous way we find the remaining links $l_{i<5}$.

A similar decomposition into colored regions occurs for arbitrary lower/upper bounds. Two edge cases are relevant. Let the range $\Delta = u - \ell$ with $u, \ell \geq 0$ the upper/lower bound. When $\Delta = 0$ we recover the $U(1)$ symmetric case discussed in the previous section, and the B, C, D regions vanish, leaving only regions A, E, F. In that case, we need to keep track of each individual QN and cannot group them. On the opposite limit, when we enlarge the bounds and $\Delta = \infty$, then the D region absorbs the other regions (A through F). This means there is actually no constraint and the corresponding MPS is the trivial product state with effectively a single segment at each link (of value $[0, \infty)$). These sanity checks reinforce the idea that the presented approach is efficient.

This approach extends to any inequality of the form $\ell \leq \sum_i a_i x_i \leq u$, even with nonpositive coefficients, where cumulative lower bounds might be negative. Each site index modifies the initial flux region, breaking it down into smaller segments as we move backward from the last tensor in the network. We will refer to these segments and the associated quantum numbers collectively as *quantum regions*, or *QRegions*. This terminology will become clearer as we explore scenarios involving multiple inequalities.

One remark which is not apparent in the inequality discussed above is that the backward decomposition sweep only gives a (super)set of *potential* QRegions. For generic inequalities a further *forward validation* sweep is needed starting from the left-most link. This guarantees that only those QRegions consistent with the relation $l_i + a_{i+1} x_{i+1} \subseteq l_{i+1}$ and the left boundary condition $l_1 \subseteq a_1 x_1$ are kept. This is because the backward decomposition sweep has only partial information about the left hand side (through the cumulative bounds), and a further

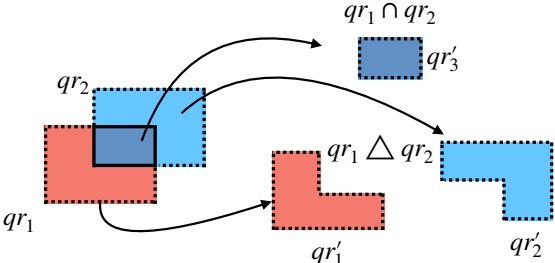

Figure 6: **Basic QRegion operations.** Given QRegions $qr_1$ and $qr_2$, shown here their intersection $qr_1 \cap qr_2$ and their symmetric difference $qr_1 \triangle qr_2$. Together they give three new QRegions, $qr_1'$, $qr_2'$, $qr_3'$.

sweep is needed to resolve those potential QRegions that could a priori appear from the backward sweep. The corresponding tensors in the MPS will be block-sparse with blocks labeled by QRegions satisfying $l_i + a_{i+1}x_{i+1} \subseteq l_{i+1}$.

Lastly, for context we compare the savings that result from our approach when compared with the slack variable approach. For concreteness we consider the inequality constraint $\sum_{i=1}^{6} x_i \leq 4$. Converting this to an equality necessarily involves the introduction of three slack variables $s_i$ so that the constraint becomes $\sum_{i=1}^{6} x_i + s_1 + 2s_2 + 4s_3 = 4$. A simple calculation reveals that the maximum set of QNs for the corresponding symmetric MPS is 5 and the total number of blocks summed over all tensors is 48. In contrast, the maximum set of QRegions in our approach is 3 (as shown in Fig. 5(b)) and the total number of blocks is 26.

## 3.2 Multiple inequalities

In the presence of $M \geq 2$ inequalities of the form $\boldsymbol{\ell} \leq \mathbf{A}\boldsymbol{x} \leq \boldsymbol{u}$ a similar logic follows to that of the one inequality case. Each QN is represented by a tuple of integers $(q_1, q_2, \cdots, q_M)$, which corresponds to a point in the $M$-dimensional hyperplane. The upper and lower constraints define a hyperrectangle or *box* $l_N = \{\boldsymbol{v} \in \mathbb{Z}^M : \ell_i \leq v_i \leq u_i\}$. This is set to be the flux of the whole MPS. The QRegions are defined by a group of QNs in the $M$-dimensional plane, or alternatively, by a disjoint union of connected regions surrounding QNs. The latter perspective will be the one we will use as it requires less bookkeeping and is more intuitive. This distinction will be illustrated in an example below.

In the backward decomposition process, the goal is to find the set of potential QRegions at each link index that recursively satisfy $l_i \subseteq l_{i+1} - A_{i+1}x_{i+1}$, subject to the cumulative bounds constraint. Here, $A_{i+1}$ is the $i+1$'th column of $\mathbf{A}$ and the operand $\subseteq$ denotes a subset relationship, ensuring that for any QRegion $q \in l_i$ there exists a QRegion $\tilde{q} \in l_{i+1} - A_{i+1}x_{i+1}$ s.t. $q \subseteq \tilde{q}$. This recursive relationship guarantees that the backward decomposition identifies all feasible QRegions while adhering to the cumulative bounds.

**Initialization algorithm.** To find the set of QRegions fulfilling the subset constraint, we introduce two operands acting on a pair of QRegions: the *intersection* $\cap$, and *symmetric difference* $\triangle$. A visual description on how they act is shown in Fig. 6 for two QRegions corresponding to boxes in $M = 2$. The same operands can be straightforwardly generalized to any set of QRegions, and in higher dimensions, $M > 2$.

The backward decomposition sweep is as follows. We assume that the flux is placed at the last site. As in the inequality case above, we first extract the cumulative upper/lower bounds as we move from left to right on the MPS and store that as a vector, $\mathcal{B}$. This is achieved by the function BOUNDARY. Next we compute the last link index, which is found by the inter-

section between the flux $\mathrm{Box}(\boldsymbol{\ell},\boldsymbol{u})$ and the cumulative upper/lower bound at the flux link, $\mathcal{B}_N$. Next, for $N-1$ iterations we extract $l_{N-i}$ from $l_{N-i+1}$ and $A_{N-i+1}$ as follows. Compute $l_{N-i}^{(0)} = l_{N-i+1} \cap \mathcal{B}_{N-i}$ and its shifted version, $l_{N-i}^{(1)} = (l_{N-i+1} - A_{N-i+1}) \cap \mathcal{B}_{N-i}$. Compute the intersection of these two indices, $l_\cap = l_{N-i}^{(0)} \cap l_{N-i}^{(1)}$, and their symmetric difference $l_\triangle = l_{N-i}^{(0)} \triangle l_{N-i}^{(1)}$. Finally, append these two outcomes to extract the new index, $l_{N-i} = l_\cap \oplus l_\triangle$. Here we overload the $\oplus$ operand to indicate that under the hood, we are decomposing a space into the disjoint union of QRegions from $l_\cap$ and $l_\triangle$, analogous to how a vector space decomposes into a disjoint sum of irreps labeled by QNs in $U(1)$ symmetric tensors.

Next we perform a forward validation sweep and discard those QRegions found on the backward decomposition sweep that are not consistent with charge conservation. Specifically, starting from the leftmost index we check whether the quantum numbers $0$, $A_1$ belong to $l_1$ and keep the corresponding QRegions to which they belong as the new $l_1$. At the following steps we check which QRegions from $l_{i+1}$ fulfill $l_i \subseteq l_{i+1}$ or $(l_i + A_{i+1}) \subseteq l_{i+1}$, and keep those as forming the new $l_{i+1}$. In this regard, it is useful to introduce the function $\chi(\cdot)$. For any pair of indices $A$ and $B$, $\chi(A \subseteq B)$ returns a new index $B$ with all QRegions in $B$ for which there exists at least one QRegion in $A$ s.t. $q_A \subseteq q_B$ with $q_B \in B$.

The series of steps involved in the backward decomposition and forward validation sweeps is captured in Algorithm 1. Next, we discuss a simple example to illustrate this algorithm.

---

**Algorithm 1** Construct MPS link indices from constraints assuming flux at rightmost site

---

**Input** Linear constraints $\{\mathbf{A}, \boldsymbol{\ell}, \boldsymbol{u}\}$
**Output** Indices $\{l_i\}$                 ($\triangleright$) Feasible solution space MPS

1: **function** CONSTRAINTSTOINDICES($\mathbf{A}, \boldsymbol{\ell}, \boldsymbol{u}$)
2:      $\mathcal{B}_{i=1,2,\cdots,N} \leftarrow \mathrm{BOUNDARY}(\mathbf{A})$          ($\triangleright$) Compute cumulative lower/upper bounds
3:      $l_N = \mathrm{Box}(\boldsymbol{\ell}, \boldsymbol{u}) \cap \mathcal{B}_N$
4:      **for** $1 \le i < N$ **do**                 ($\triangleright$) Backward decomposition sweep
5:          $l_{N-i}^{(0)} \leftarrow l_{N-i+1} \cap \mathcal{B}_{N-i}$
6:          $l_{N-i}^{(1)} \leftarrow (l_{N-i+1} - A_{N-i+1}) \cap \mathcal{B}_{N-i}$
7:          $l_\cap \leftarrow l_{N-i}^{(0)} \cap l_{N-i}^{(1)}$
8:          $l_\triangle \leftarrow l_{N-i}^{(0)} \triangle l_{N-i}^{(1)}$
9:          $l_{N-i} \leftarrow l_\cap \oplus l_\triangle$
10:      **end for**
11:      $l_1 \leftarrow \chi(0 \subseteq l_1) \oplus \chi(A_1 \subseteq l_1)$
12:      **for** $1 \le i < N$ **do**                 ($\triangleright$) Forward validation sweep
13:          $l_{i+1} \leftarrow \chi(l_i \subseteq l_{i+1}) \oplus \chi((l_i + A_{i+1}) \subseteq l_{i+1})$
14:      **end for**
15:      **return** $\{l_i\}$
16: **end function**

---

**Example.** To illustrate the initialization algorithm for multiple inequalities, we consider the following two inequalities:

$$-1 \le x_1 + 2x_2 - x_3 - 2x_4 \le 2,$$
$$-1 \le -2x_1 + 3x_2 - x_3 + x_4 \le 1. \tag{5}$$

The corresponding constrained MPS is depicted in Fig. 7, where, as in Fig. 4, we've denoted by red/blue numbers the cumulative lower/upper bounds of charge on $l_i$ w.r.t. each inequality. The backward decomposition sweep is illustrated in Fig. 8. It starts from the flux of the MPS at the last tensor, given by $l_4 = [(-1,-1),(2,1)]$, which is a

rectangle defined by the coordinates of the down-left and top-right corner in the format $[(x_{1,\min}, x_{2,\min}),(x_{1,\max}, x_{2,\max})]$, plotted as a green rectangle in Fig. 8(a). We define $Q_i$ as the number of QRegions in $l_i$. So $Q_4 = 1$ since it contains only one QRegion. The link index $l_3$ is found by the intersection and symmetric differences from lines 7, 8 of Algorithm 1, with the boundary $\mathcal{B}_3 = [(-1,-3),(3,3)]$. This results in $l_\cap = [(1,-1),(2,0)]$ and $l_\triangle = \{[(-1,-1),(0,0)] \cup [(-1,1),(2,1)], [(1,-2),(3,-2)] \cup [(3,-1),(3,0)]\}$. Using line 9 of Algorithm 1 we get

$$
\begin{aligned}
l_3 &= l_\cap \oplus l_\triangle \\
&= \{[(1,-1),(2,0)], [(-1,-1),(0,0)] \cup [(-1,1),(2,1)], [(1,-2),(3,-2)] \cup [(3,-1),(3,0)]\},
\end{aligned}
$$

which is a link index with three QRegions, thus $Q_3 \le 3$. The upper bound in $Q_3$ being since some of these QRegions may ultimately not appear upon implementing the forward validation sweep. Visually $l_3$ corresponds to Fig. 8(b) where the different QRegions are shaded in yellow, blue (symmetric difference) and green (intersection). We note that QRegions such as the L-shape in yellow or blue can be represented as the union of disjoint boxes. This is in fact how we encode arbitrary QRegions in code.

The remaining indices can be found in an analogous manner yielding visually Fig. 8(c-d). From (c) we see that $l_2$ consists of multiple pieces, most of which contains only one QN in its QRegion. For ease of exposition, we call these QNs as QRegions. These *singleton* QRegions are not colored for simplicity in (c) or (d). So $l_2$ contains three *regular* and nine singleton QRegions (QNs), with a total of twelve QRegions.

Once the backward process is over the next step is to proceed with the forward validation process, lines 11-14 in Algorithm 1. Starting from the boundary condition $l_1 = A_1 x_1$, we check which QRegions in $l_{i+1}$ fulfill $l_i + A_{i+1} x_{i+1} \subseteq l_{i+1}$. This process selects the QRegions shown in Fig. 9(a-d), with max $Q_i = 3$. The black dots correspond to the QNs that would be selected had we instead solved $\hat{l}_{i+1} = \hat{l}_i + A_{i+1} x_{i+1}$ subject to $\hat{l}_{i+1} \subseteq l_{i+1}$ with $\hat{l}_1 = A_1 x_1$. In that case the number of QNs at site 3 and 4 are $Q_3 = Q_4 = 4$ which is greater than the maximum number of QRegions. Hence, by working with QRegions we guarantee the most compact representation of a constrained MPS in terms of bond dimension. While QRegions may include QNs that are not part of the solution space (e.g., $(0,-1)$ at link index $l_3$), this does not increase the bond dimension of the MPS. We refer to them as non-valid QNs not because they violate constraints, but because they cannot be realized by any bit string. For example, in Fig. 8(a), for $l_4$, all 12 QNs satisfy all constraints; however, only 4 of them can be realized in Fig. 9(d). This means that there is no valid solution corresponding to certain QNs. The inclusion of such *virtual* QNs in QRegions is a practical choice to simplify the description of QRegions — for example, instead of considering $l_4$ as the union of the four black dots in Fig. 9(d), it is simpler to consider the green rectangle in its entirety, thus storing only two coordinates for the corners of the rectangle, as opposed to four for the dots. The question that remains open is whether representing each QRegion as a union of individual QNs, versus as a contiguous region (which may include invalid QNs), leads to advantages in terms of memory layout or computational efficiency.

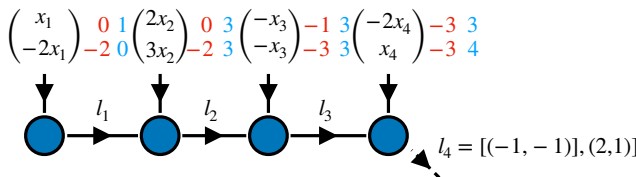

Figure 7: **Constrained MPS with flux at last site.** Corresponding to Eq. (5).

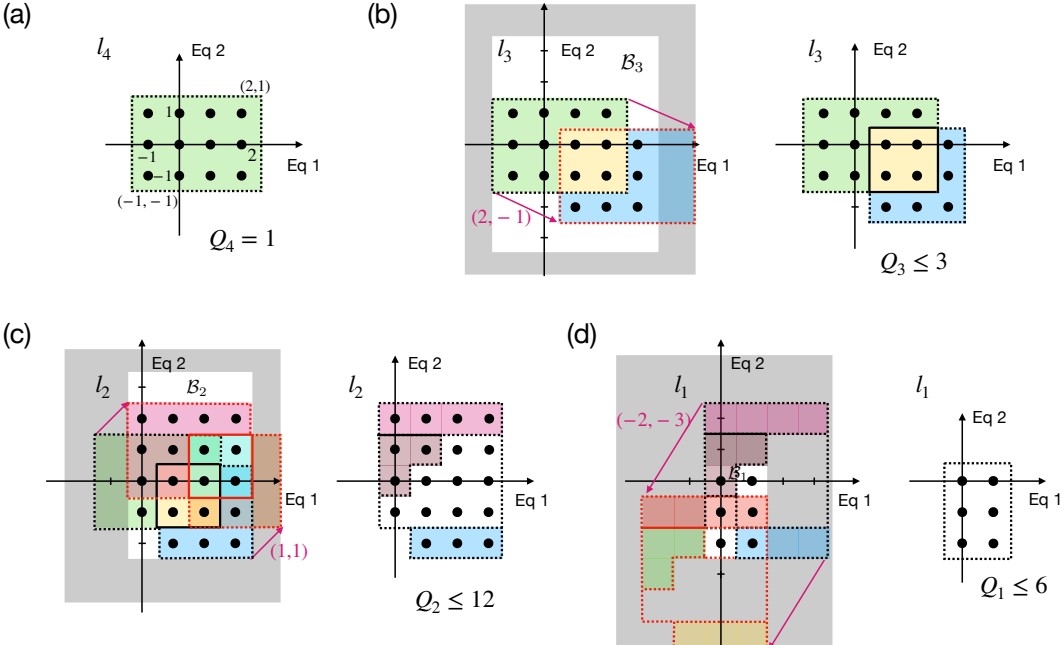

Figure 8: **Backward decomposition sweep.** Here shown for the two inequalities of Eq. 5. For each link index $l_i$, $Q_i$ denotes the number of feasible QRegions. This number is upper bounded by the number of QRegions identified during the backward sweep. *(a)* The feasible QRegion of $l_4$ is given by the green area. *(b,c,d)* QRegions at link indices $l_3$, $l_2$, $l_1$, respectively. Each colored region represents a distinct QRegion. Uncolored dots denote singleton QRegions (i.e., individual QNs) omitted from coloring for clarity. The gray shaded area in each panel marks values beyond the feasible boundary $B_i$ of possible QRegions/QNs. The pink arrow is given by $-A_i$, from step 5 of Algorithm 1.

Crucially, although QRegions may include non-valid QNs, the fusion rules governing the tensor contractions guarantee that only feasible global configurations contribute to the MPS. This can be verified for our example

$$\psi[x_1, x_2, x_3, x_4] = T_{l_1}^{1(x_1)} T_{l_1, l_2}^{2(x_2)} T_{l_2, l_3}^{3(x_3)} T_{l_3}^{4(x_4)} \neq 0 \Leftrightarrow (x_1, x_2, x_3, x_4) \in \mathcal{S},$$

where $\mathcal{S}$ denotes the feasible space. Thus, no spurious solutions are sampled or optimized over, even if individual QRegions include QNs that are never visited. Conversely, all feasible solutions yield a nonzero contraction, ensuring full coverage of the feasible set.

The above analysis assumed that the flux was placed at the last tensor. Moving the flux center in $U(1)$ MPS is straightforward and can be done dynamically by solving the equality constraint $qn_{\text{left}} + qn_{\text{right}} + \alpha x = \text{flux}$, with $\alpha \in \mathbb{Z}$ and $x \in \{0, 1\}$ the site index QN. In particular, if we move the flux tensor from site $i$ to site $i + 1$, we only need to solve for $qn_{\text{left}}$; see Sec. 2. In contrast, for constrained tensors involving arbitrary QRegions, the flux condition becomes $qr_{\text{left}} + qr_{\text{right}} + \alpha x \subseteq \text{flux}$. This condition is ambiguous if we were to solve say for the set of QRegions on the left index. In order to determine it, we need to find the QRegions when fixing the flux at the first site, which results in the MPS of Fig. 10.

Letting $\tilde{l}_i$ the set of link indices resulting from placing the flux at the first site and applying Algorithm 1 with **A** reversed (so that $A_i \to A_{N-i+1}$), yields the following QRegions

$$\tilde{l}_1 = \{[(-2,2),(1,3)] \cup [(-2,1),(-2,1)], [(-1,-1),(2,0)] \cup [(2,1),(2,1)]\},$$
$$\tilde{l}_2 = \{[(0,-1),(0,0)], [(-1,-1),(-1,0)], [(-2,1),(-2,1)], [(-3,-1),(-2,0)]\},$$
$$\tilde{l}_3 = \{[(0,0),(0,0)], [(-2,1),(-2,1)]\}.$$

Note in particular that $4 = |\tilde{l}_2| \neq |l_2| = 3$, and $2 = |\tilde{l}_3| \neq |l_3| = 3$.

Away from the edges, the flux tensor will have blocks labeled by QRegions fulfilling $l_{i-1} + \tilde{l}_i + A_i x_i \subseteq \text{flux} = [(-1,-1),(2,1)]$, where the addition of QRegions is understood as follows. For two boxes appearing on each QRegion $\text{box}_1 = [(x_{1,\min},x_{2,\min}),(x_{1,\max},x_{2,\max})]$ from QRegion 1 and $\text{box}_2 = [(y_{1,\min},y_{2,\min}),(y_{1,\max},y_{2,\max})]$ from QRegion 2, their addition is done component wise, thus

$$\text{box}_1 + \text{box}_2 = [(x_{1,\min}+y_{1,\min},x_{2,\min}+y_{2,\min}),(x_{1,\max}+y_{1,\max},x_{2,\max}+y_{2,\max})].$$

Figure 9: **Forward validation sweep.** Here shown for the two inequalities of Eq. 5. *(a-d)* QRegions extracted in the forward validation sweep solving using lines 11-14 of Algorithm 1 The dots denote the QNs within each QRegion that are actually selected – see main text. *(e)* The block-sparse tensor structures for all tensors in the MPS. Each column and row are labeled by its corresponding QRegion. The meaning of the color and shape is the same as in panels *(a-d)*. The symbol $X$ is a placeholder for a tensor block.

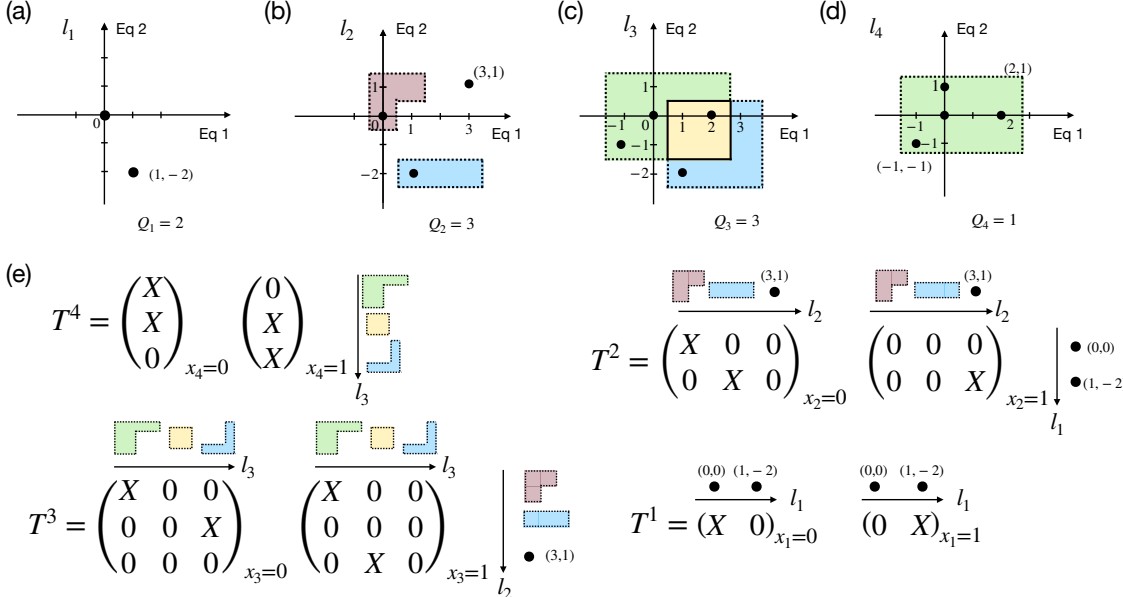

Figure 10: **Constrained MPS with flux at first site.** Corresponding to Eq. (5).

Fixing the flux at site $m$, we can construct the MPS tensors as in Algorithm 2. For simplicity we set all nonzero blocks to the scalar value 1, corresponding to a uniform superposition of all feasible solutions to the associated set of constraints.

One striking consequence of constrained tensors labeled by QRegions is that in contrast to the $U(1)$ symmetric and vanilla cases (no constraints), the bond dimension at the last link can be of value 3 since the flux $l_N = \text{Box}(\ell, u)$ and its shift by the last site index can be nonempty, $l_N \cap (l_N - A_N) \neq \varnothing$. This can produce three new QRegions as in Fig. 6 for the last link.

---

**Algorithm 2** Construct MPS from constraints

**Input**     Linear constraints $\{A, \ell, u\}$, flux site $m$
**Output**     MPS $\psi$                               ($\triangleright$) Feasible solution space MPS
 1: **function** CONSTRAINTSTOMPS($A, \ell, u$)
 2:      $\{l_i\} \leftarrow$ CONSTRAINTSTOINDICES($A, \ell, u$)        ($\triangleright$) Extract MPS link indices with flux at rightmost site
 3:      $\{\tilde{l}_i\} \leftarrow$ CONSTRAINTSTOINDICES(reverse($A$), $\ell, u$)     ($\triangleright$) Extract MPS link indices with flux at leftmost site
 4:      **for** $i < m$ **do**
 5:          $T^{i(x_i)}_{l_{i-1}, l_i} \leftarrow 1$                        ($\triangleright$) Initialize tensor components satisfying $l_{i-1} + A_i x_i \subseteq l_i$
 6:      **end for**
 7:      **for** $m < i < N$ **do**
 8:          $\tilde{T}^{i(x_i)}_{\tilde{l}_{i-1}, \tilde{l}_i} \leftarrow 1$                        ($\triangleright$) Initialize tensor components satisfying $\tilde{l}_i + A_i x_i \subseteq \tilde{l}_{i-1}$
 9:      **end for**
10:      $\overline{T}^{m(x_m)}_{l_{m-1}, \tilde{l}_m} \leftarrow 1$          ($\triangleright$) Initialize flux tensor with components satisfying $\tilde{l}_m + l_{m-1} + A_m x_m \subseteq \text{Box}(\ell, u)$
11:      **return** $\psi = T^1_{l_1} T^2_{l_1, l_2} \cdots T^{m-1}_{l_{m-2}, l_{m-1}} \overline{T}^m_{l_{m-1}, \tilde{l}_m} \tilde{T}^{m+1}_{\tilde{l}_m, \tilde{l}_{m+1}} \cdots \tilde{T}^{N-1}_{\tilde{l}_{N-2}, \tilde{l}_{N-1}} \tilde{T}^N_{\tilde{l}_{N-1}}$
12: **end function**

---

# 4 Complexity of constrained tensor networks

## 4.1 Complexity of algorithm 1

Having discussed the construction of constrained tensor networks for arbitrary linear constraints we study now the complexity of Algorithm 1. The bottleneck of the algorithm occurs at steps 7 and 8. Step 7 finds the intersections among all pairs of QRegions appearing on indices $l^{(0)}_{N-i}$ and $l^{(1)}_{N-i}$. Each QRegion is in turn composed of a disjoint union of boxes. Thus, at the most basic stack one performs the intersection of two boxes, which scales as $\mathcal{O}(M)$, with $M$ the number of constraints. Step 8 is also dominated by the intersection of pairs of QRegions. The number of such intersections will be ultimately determined by the specific constraints. In general, we do observe that the more structure there is in the constraints (as measured e.g. by the variance of coefficients), the fewer QRegions there will be. The dependence of the intersection step on the number of constraints has an impact on the complexity of the algorithm. The natural measure of complexity in this setup is the maximum number of QRegions across all tensors, and among both the MPS with flux at either end. We dub this quantity *charge complexity*, denoted by $Q$.

Note that each QRegion could be composed of multiple boxes, which adds complexity, but since the rank of each tensor is determined by the number of QRegions, we choose this quantity instead. In general, we observe that for well-structured dense constraint matrices $A$ (i.e., global constraints with e.g. low variance in coefficients), the scaling of $Q$ is exponential in $M$ (for fixed $N$) but at most polynomial in the number of bits $N$ (for fixed $M$). This exponential dependence of bond dimension on the number of constraints $M$ was also observed in [14] for a 2SAT problem in CNF. The dependence on both $N$ and $M$ is illustrated below for a family of constraints that arise in a version of the facility location problem.

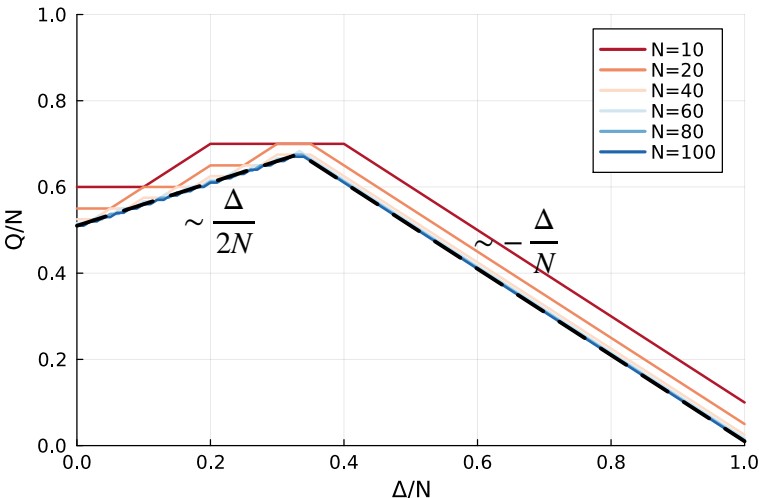

Figure 11: **Charge complexity for cardinality constraint.** Scaling collapse of charge complexity vs range for cardinality constraint. The maximum complexity occurs at a value at $\Delta^* = N/3$ with complexity $Q(\Delta^*) = 2N/3$. At $\Delta = 0$, our approach recovers the charge complexity of $Q = N/2 + 1$ expected for cardinality equality constraint. At $\Delta = N$, we recover the expected charge complexity of $Q = 1$, corresponding to a product state (bond dimension 1). In between we observe only mild linear growth of $Q$ in $\Delta$. This suggests our encoding in terms of QRegions is near optimal.

## 4.2 Charge complexity of constrained MPS

**Charge complexity for cardinality constraint.** While we do not have guarantees that our constrained tensor network construction is optimal, we give an intuition that this is so for a simple constraint, given by $\ell \le \sum_i x_i \le u$. Suppose we want to describe the TN corresponding to an inequality with no lower or upper bounds (i.e. $\ell = -\infty$ and $u = \infty$, for lower and upper bounds, respectively). The smallest rank TN representation for this problem, corresponding to setting all TN blocks to be of size one, is given by a trivial product state. When $u = \ell = N/2$, the charge complexity is given by $Q = N/2 + 1$ [10]. In between these two limits the charge complexity scales linearly with the range $\Delta = u - \ell$. In Fig. 11 we show this for a specific set of $\Delta$ parameterized as $\Delta_i = u_i - \ell_i$, $i = 0, 1, 2, \cdots$, with $u_i = N/2 + i/2$, $\ell_i = N/2 - i/2$. In Fig. 11 we show via a scaling collapse that the complexity as a function of $N$ for this particular example scales linearly as $Q = \alpha N$, with $\alpha < 1$. In fact, we can derive the large $N$ limit case with the help of Fig. 5(c). The charge complexity is given by $Q = N - \Delta$ when $\Delta \ge \frac{1}{3}N$, $Q = \frac{N+\Delta}{2}$ when $\Delta \le \frac{1}{3}N$. The maximum complexity as a function of the range $\Delta$ corresponds to $Q(\Delta^*) = \frac{2}{3}N$, with $\Delta^* = \frac{1}{3}N$.

**Charge complexity for facility location problem.** The facility location problem is a classic optimization problem in operations research and supply chain management. The goal is to determine the most cost-effective locations for facilities (e.g., warehouses, factories, retail stores) and to allocate demand points (e.g., customer locations, market areas) to these facilities to minimize overall costs while satisfying service requirements. Here we model the set of constraints appearing in this problem by a set of inequality constraints of the form

$$\ell \le \sum_{j \in \text{Facilities}} x_{i,j} \le u, \qquad i = 1, 2, \ldots, M. \tag{6}$$

Here $M$ is the number of demand points. For concreteness, we fix $\ell = 2$, i.e. two facilities must serve each demand point at any given time, and let the maximum number of facilities

per demand point $u$ vary. Furthermore, each demand point is served by 10% of the facilities, chosen randomly. All in all, the constraint matrix $A$ is an $M \times N$ matrix whose entries are in $\{0, 1\}$ such that $\sum_j A_{i,j} = \lfloor 0.1N \rfloor$. The locations of $A_{i,j} = 1$ are randomly chosen.

In Fig. 12 we show the charge complexity as a function of the number of bits $N$ for different numbers of constraints $M$ for $u = 2$ (minimum), $u = 3$, and $u = 10$ (maximum). For each tuple of $(N, M, u)$ we construct 5 different constraint matrices $A$ as above and extract the statistics of the charge complexity (mean and standard error), remarking that the results are not very sensitive to the specific choice of matrix $A$.

The results show that, for fixed $M$ and $u$, the charge complexity scales at most polynomially with $N$, potentially linearly, although the exact polynomial dependence remains unclear. In particular, complexity saturates at large $N$. Interestingly, the results for the minimum ($u = 2$) and maximum ($u = 10$) number of facilities per demand point exhibit similar charge complexities. This highlights the efficiency of our constrained tensor network ansatz in handling inequality constraints over a large range, contrasting with the slack variable method, where complexity increases with the range due to the addition of slack bits.

We also show that the charge complexity increases exponentially as a function of the number of constraints $M$ for fixed $N$ and $u$. Finally we show the charge complexity as a function of the upper bound $u$ for fixed $N$ and $M$. The behavior is consistent with that of the cardinality example discussed earlier: the complexity increases polynomially up to some range $\Delta = u - \ell$, and decreases polynomially beyond that point. As before, we cannot rule out a linear dependence in $u$.

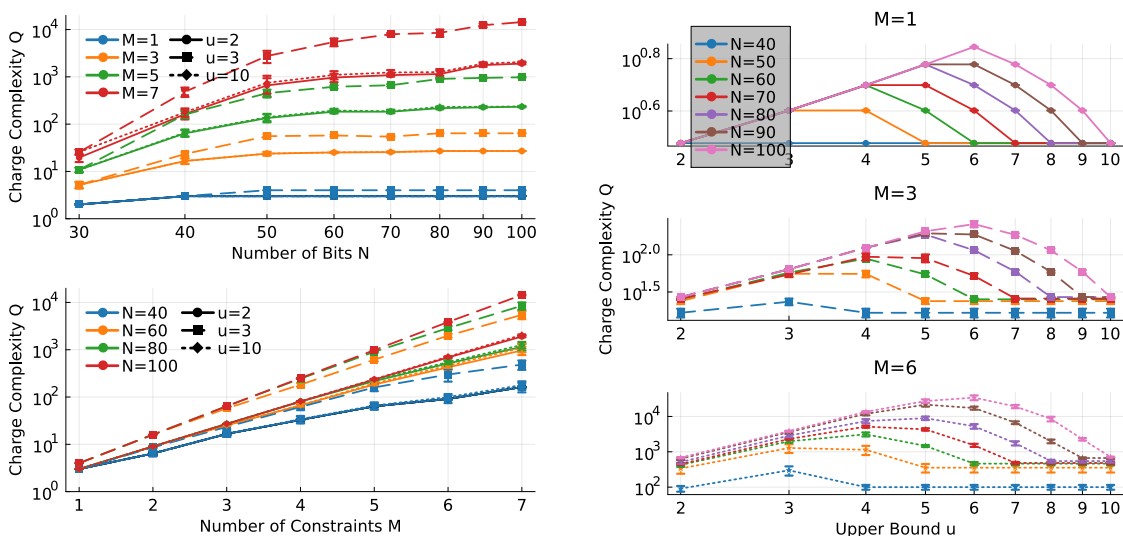

Figure 12: **Charge complexity for facility constraints.** Top left panel illustrates the charge complexity as a function of the number of bits $N$ for varying numbers of constraints $M$ with upper bounds $u$ set at 2 (minimum), 3, and 10 (maximum). We present data for 5 different constraint matrices $A$ per tuple $(N, M, u)$, showing mean charge complexity and standard error; the low sensitivity to $A$ is evident as error bars are barely visible. The charge complexity generally exhibits polynomial scaling with $N$ for a fixed set of constraints. Bottom left panel shows the charge complexity as a function of number of constraints $M$ for fixed $N$ and $u$, showing exponential scaling. Right panel examines charge complexity as a function of the upper bound $u$ for constant $N$ and $M$, reflecting a polynomial increase up to a specific range $\Delta = u - \ell$, followed by a polynomial decrease. This trend parallels the cardinality results discussed previously.

### 4.3 Complexity of contracting the constrained MPS

An important consequence of the tensor-network formulation described above is that contracting the constrained MPS directly provides the number of feasible solutions of the underlying system of inequalities, provided that we set all nonzero tensor blocks to be of value 1 (i.e., blocks of size 1), as in Algorithm 2. This yields $\psi(\boldsymbol{x}) = 1$ for any feasible bitstring $\boldsymbol{x}$. Denoting the feasible solution space by $\mathcal{S} = \{\boldsymbol{x} \in \{0,1\}^N : \boldsymbol{\ell} \leq \mathbf{A}\boldsymbol{x} \leq \boldsymbol{u}\}$, the total number of feasible solutions is

$$|\mathcal{S}| = \sum_{\boldsymbol{x} \in \mathcal{S}} \psi(\boldsymbol{x}) = \sum_{\boldsymbol{x} \in \{0,1\}^N} T_{l_1}^{1(x_1)} T_{l_1,l_2}^{2(x_2)} \cdots T_{l_N}^{N(x_N)}. \tag{7}$$

The complexity of evaluating the right-hand side via tensor network contraction will depend on the resulting block-sparsity of each tensor. Assuming there are only $\mathcal{O}(Q)$ tensor blocks (of size 1) on every tensor (this holds in particular for equality constraints of cardinality type where blocks lie along a diagonal), the scaling is simply $\mathcal{O}(NQ)$.

## 5 Canonical form and compression

Section 3 discussed how to find the set of QRegions for an arbitrary set of inequalities. These will be the labels of each block that appear on each tensor in the MPS. A nice property about MPS in general is that they afford a canonical form [12]. This fixes the *gauge* degree of freedom that arises from equivalent representations of the same MPS, a consequence of the fact that inserting any pair of invertible matrices $P$ and $P^{-1}$ at any link of the MPS leaves the resulting MPS invariant. Among the different choices of gauge, there exists a particular one so that the optimal truncation of the global MPS can be done directly on a single tensor, the *canonical tensor*.

Canonical forms exist for vanilla and $\mathcal{G}$ symmetric MPS (with $\mathcal{G}$ some arbitrary local or global symmetry group). Here we will show that constrained MPS afford a canonical form as well. Recall that an MPS is canonicalized with canonical center at site $i$ if all tensors to the left of this are *left isometries*, $\sum_{j=0,1}(T^{(j)})^\dagger T^{(j)} = \mathbb{1}$, and all tensors to the right are *right isometries*, $\sum_{j=0,1} T^{(j)}(T^{(j)})^\dagger = \mathbb{1}$. Such canonical forms can always be accomplished via a series of orthogonal factorizations such as QR or SVD, leaving the *nonorthogonal* tensor as the canonical tensor, and the rest being isometries.

In order to canonicalize one must be able to matricize each rank-3 tensor in the MPS, via fusing each site index with one of the link indices, followed by splitting after factorization is achieved. Fusing and splitting of indices can be done on $\mathcal{G}$ symmetric tensors via use of Clebsh-Gordan coefficients. For $\mathcal{G} = U(1)$ such coefficients become straightforward and amount to solving linear equations of the form $n_1 + n_2 = n_{12}$, with $n_{12}$ the merged index. For constrained tensors arising from inequality constraints fusion and splitting can't be done in an analogous manner, as each subspace corresponding to a block is labeled by a QRegion and flux conservation dictates $qr_1 + qr_2 \subseteq qr_{12}$. Such an undetermined system requires a different strategy.

The approach taken here is to precompute all link indices by setting the flux of the MPS at the first and the last sites, as explained before. This will determine all indices $\{l_i\}$ and $\{\tilde{l}_i\}$. Suppose we have canonicalized the MPS to be at site $i$. It suffices to show how to shift the canonical center from site $i$ to site $i+1$, as depicted in Fig. 13. Here the canonical tensor is denoted in green, which is a *flux matrix* $F$ defined between $i$ and $i+1$. All tensors in blue on the left $T^1, T^2, \cdots, T^i$ are the same tensors appearing in the left canonical MPS, while those in orange on the right $\tilde{T}^{i+1}, \tilde{T}^{i+2}, \cdots, \tilde{T}^N$ tensors are the same ones appearing in the right

canonical MPS. These tensors satisfy the following condition

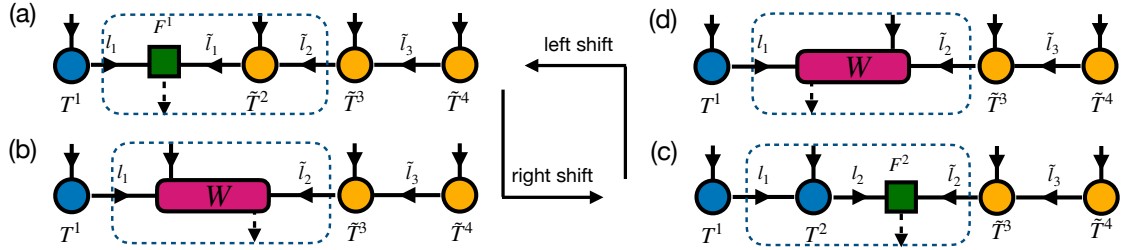

In Fig. 13, we show the whole process of shifting the canonical center from $F^1$ to $F^2$. We first contract the $F^1$ and $\tilde{T}^2$ tensor into $W$ tensor in (b). Then we factorize and truncate into new tensors $T^2$ and $F^2$ in (c). The updates only happens locally in the dashed region, which can be captured by the conditions

$$F^{i-1}\tilde{T}^i = T^i F^i \,, \tag{8}$$

$$\text{subject to } F^0 = F^{N+1} = \mathbb{1} \,, \tag{9}$$

$$\sum_j \tilde{T}^{i(j)}\left(\tilde{T}^{i(j)}\right)^\dagger = \mathbb{1}, \, l_{i-1} + \tilde{l}_{i-1} \subseteq \phi \,, \tag{10}$$

$$\sum_j \left(T^{i(j)}\right)^\dagger T^{i(j)} = \mathbb{1}, \, l_i + \tilde{l}_i \subseteq \phi \,, \tag{11}$$

with $\phi$ the flux. These conditions are on top of the ones discussed earlier for each of the tensors away from the flux, i.e. $\tilde{l}_i + A_i x_i \subseteq \tilde{l}_{i-1}$ and $l_{i-1} + A_i x_i \subseteq l_i$. The upshot is that these constraints allow for the following features: 1) The number of blocks at a given tensor is not necessarily preserved when we move the canonical center; i.e. the number of blocks in $T^i$ and $\tilde{T}^i$ may differ. 2) Relatedly, factorizations may occur on joint blocks. This is a consequence of the fact that two merged indices may have multiple QRegions contained in a given QRegion of a third index. 3) In order to determine which blocks to factorize jointly we need to supplement the factorization with information from the new index that will result from merging. This may incur an increase in complexity, as shown in Fig. 11.

Figure 13: **Canonicalization of an MPS.** The *(a)-(b)-(c)* panels explain the right shift of canonical center while *(c)-(d)-(a)* panels show the left shift of the canonical center. *(a)* The MPS canonical form. The MPS contains three parts. The green tensor (matrix) is the canonical center defined on link between 1st and 2nd site, which carries the flux. All the blue tensors are the same tensors that appear on the left canonical MPS, while the orange tensors are from the right canonical MPS. *(b)* If we want to move the canonical center to the right, we first contract $F^1$ and $\tilde{T}^2$ and get $W$. *(c)* Then we try to factorize $W$ and get $T^2$ and $F^2$. At this stage we have successfully moved the canonical center one site to the right. *(d)* Shows the intermediate state when left shifting the canonical center. The $W$ tensor is the same as in *(b)*, however, it is reshaped into a different matrix (see main text).

The three index tensor $W \in \mathbb{R}^{\chi_L \times d \times \chi_R}$ can be matricized by merging the site index with one of the link indices. Considering first the merging with the left index we get $W \in \mathbb{R}^{d\chi_L \times \chi_R}$, which can then be used to factorize, and if necessary, compress via singular value decomposition (svd). Factorization results in the product of $\bar{U} \in \mathbb{R}^{d\chi_L \times \chi}$ and $F \in \mathbb{R}^{\chi \times \chi_R}$. The new tensor $T^2$ would be the reshape of $\bar{U}$ here:

$$
\underset{\substack{W \in \mathbb{R}^{d\chi_L \times \chi_R}}}{\overset{d \quad \tilde{l}_2}{\underset{\chi_L \qquad \chi_R}{\longrightarrow}}} \approx \underset{\bar{U} \in \mathbb{R}^{d\chi_L \times \chi} \quad F \in \mathbb{R}^{\chi \times \chi_R}}{\overset{d\chi_L \quad \chi \quad \chi_R}{\underset{l_2}{\longrightarrow}}} \; .
$$

Notably, $\bar{U}$ is not a square matrix after truncation. $\chi \leq \min(d\chi_L, \chi_R)$ is the dimension maintained after truncation. $\bar{U}$ does not carry flux, i.e. we have QRegion conservation and it corresponds to a block diagonal matrix and can be written as $\bar{U} = \oplus_{i=1}^{q} \bar{U}_i$, where $\bar{U}_i \in \mathbb{R}^{D_i \times \bar{D}_i}$ represent different blocks. The row dimension is not truncated $\sum_i D_i = \chi_L d$, while the column dimension satisfies $\sum_i \bar{D}_i = \chi$. The orthogonal condition of $\bar{U}$ would require that each $\bar{U}_i$ would be an isometry. Our goal is to factorize $W$ so that the norm-2 error is as minimal as possible subject to the canonical condition of $\bar{U}$ in Eq. 11:

$$
\bar{U}_i, \bar{F} = \mathrm{argmin}\left( \left\| W - \bar{U}\bar{F} \right\|^2 \right), \quad \text{s.t.} \quad \bar{U}_i^\dagger \bar{U}_i = I_i, \quad \forall i. \tag{12}
$$

For convenience, we use symbol with bars to represent truncated matrix or tensors in our discussion.

We will give the solution of the optimization directly in the following. The proof is provided in Appendix A.

**Solution.** We split $W \in \mathbb{R}^{d\chi_L \times \chi_R}$ along the row dimension, and get the vertical concatenation of $W_i \in \mathbb{R}^{D_i \times \chi_R}$:

$$
W = \begin{pmatrix} W_1 \\ W_2 \\ \vdots \\ W_q \end{pmatrix}. \tag{13}
$$

The division of the row dimension depends on row of $\bar{U}$. Then we factorize each $W_i$ by svd, and get

$$
W_i = \bar{U}_i \bar{\Lambda}_i \bar{V}_i^\dagger, \tag{14}
$$

where $\bar{U}_i \in \mathbb{R}^{D_i \times \bar{D}_i}$, $\bar{\Lambda}_i \in \mathbb{R}^{\bar{D}_i \times \bar{D}_i}$ is diagonal matrix, $\bar{V}_i \in \mathbb{R}^{\chi_R \times \bar{D}_i}$ is a tall matrix in general. The decomposition happens independently, however, the truncation threshold is considered across all blocks. For example, before truncation, we use $\Lambda_i = (\lambda_i^1, \lambda_i^2, \cdots, \lambda_i^{D_i})$ to denote singular values before truncation for each $i$, we then sort different $\lambda_i^j$ in a descending order, and keep the largest $\chi$ values. Only the corresponding dimensions are kept during the truncation. In general, the accept ratio is dynamically adjusted based on different weights $\Lambda_i$.

The solution $T^i$ with $i = 2$ in Fig. 13(c) would be the reshape of $\bar{U}$ given by

$$
\bar{U} = \oplus_{i=1}^{q} \bar{U}_i, \tag{15}
$$

$$
T^i = \mathrm{reshape}(\bar{U}, [\chi_L, d, \chi_\chi]). \tag{16}
$$

And the $\bar{F}$ would be given by the vertical concatenation format as

$$
\bar{F} = \begin{pmatrix} \bar{\Lambda}_1 \bar{V}_1^\dagger \\ \bar{\Lambda}_2 \bar{V}_2^\dagger \\ \vdots \\ \bar{\Lambda}_q \bar{V}_q^\dagger \end{pmatrix}. \tag{17}
$$

**Example.** We will illustrate the shift of canonical center with the example of two inequalities (5), and show the steps from Fig. 13.

**Right shifting of canonical center:** Assume we're given $F^1$ and $\tilde{T}^2$. Our goal is to determine $F^2$. We can calculate $W$ and get

$$
W = \begin{pmatrix} A & 0 & B & 0 \\ 0 & C & 0 & 0 \\ 0 & 0 & 0 & 0 \\ 0 & 0 & D & E \end{pmatrix}. \tag{18}
$$

We can split it into merged *row blocks* $W_i$ with $i = 1, 2, 3, 4$, and factorize them:

$$
W_1 = \begin{pmatrix} A & 0 & B & 0 \end{pmatrix} = \bar{U}_1 \bar{\Lambda}_1 \bar{V}_1^\dagger, \tag{19}
$$

$$
W_2 = \begin{pmatrix} 0 & C & 0 & 0 \end{pmatrix} = \bar{U}_2 \bar{\Lambda}_2 \bar{V}_2^\dagger, \tag{20}
$$

$$
W_4 = \begin{pmatrix} 0 & 0 & D & E \end{pmatrix} = \bar{U}_4 \bar{\Lambda}_4 \bar{V}_4^\dagger. \tag{21}
$$

We omit $W_3$ because it's zero matrix. Based on the block structure of $W_i$, we know that the $\bar{V}_i^\dagger$ has the structure

$$
\bar{V}_1^\dagger = \begin{pmatrix} \bar{X}_A & 0 & \bar{X}_B & 0 \end{pmatrix}, \tag{22}
$$

$$
\bar{V}_2^\dagger = \begin{pmatrix} 0 & \bar{X}_C & 0 & 0 \end{pmatrix}, \tag{23}
$$

$$
\bar{V}_4^\dagger = \begin{pmatrix} 0 & 0 & \bar{X}_D & \bar{X}_E \end{pmatrix}. \tag{24}
$$

We can simplify the SVD of $W_i$ by ignoring the zero entries. For example,

$$
\text{svd}\left(\begin{pmatrix} A & B \end{pmatrix}\right) = \bar{U}_1 \bar{\Lambda}_1 \begin{pmatrix} \bar{X}_A & \bar{X}_B \end{pmatrix}, \tag{25}
$$

where $\bar{U}_1$, $\bar{\Lambda}_1$, $\bar{X}_A$ and $\bar{X}_B$ corresponding to those in Eq. (22). The $\bar{U}_1$ is extracted from a joint SVD decomposition by concatenating $A$ and $B$ block. Similarly we now have

$$
\text{svd}\left(\begin{pmatrix} C \end{pmatrix}\right) = \bar{U}_2 \bar{\Lambda}_2 \begin{pmatrix} \bar{X}_C \end{pmatrix}, \tag{26}
$$

$$
\text{svd}\left(\begin{pmatrix} D & E \end{pmatrix}\right) = \bar{U}_4 \bar{\Lambda}_4 \begin{pmatrix} \bar{X}_D & \bar{X}_E \end{pmatrix}. \tag{27}
$$

Finally we accomplish the factorization as

$$
W = \begin{pmatrix} A & 0 & B & 0 \\ 0 & C & 0 & 0 \\ 0 & 0 & 0 & 0 \\ 0 & 0 & D & E \end{pmatrix} = \begin{pmatrix} \bar{U}_1 & 0 & 0 \\ 0 & \bar{U}_2 & 0 \\ 0 & 0 & 0 \\ 0 & 0 & \bar{U}_4 \end{pmatrix} \times \begin{pmatrix} \bar{\Lambda}_1 \bar{X}_A & 0 & \bar{\Lambda}_1 \bar{X}_B & 0 \\ 0 & \bar{\Lambda}_2 \bar{X}_C & 0 & 0 \\ 0 & 0 & \bar{\Lambda}_3 X_D & \bar{\Lambda}_3 \bar{X}_E \end{pmatrix},
$$

and extract $T^2$

$$
T^2 = \begin{pmatrix} \bar{U}_1 & 0 & 0 \\ 0 & \bar{U}_2 & 0 \end{pmatrix}_{x_2=0} \begin{pmatrix} 0 & 0 & 0 \\ 0 & 0 & \bar{U}_4 \end{pmatrix}_{x_2=1},
$$

and

$$
F^2 = \begin{pmatrix} \bar{\Lambda}_1 \bar{X}_A & 0 & \bar{\Lambda}_1 \bar{X}_B & 0 \\ 0 & \bar{\Lambda}_2 \bar{X}_C & 0 & 0 \\ 0 & 0 & \bar{\Lambda}_3 X_D & \bar{\Lambda}_3 \bar{X}_E \end{pmatrix}.
$$

The $T^2$ can be verified to satisfy the canonical condition

$$
\sum_i (T^{2(i)})^\dagger T^{2(i)} = \begin{pmatrix} \bar{U}_1^\dagger \bar{U}_1 & & \\ & \bar{U}_2^\dagger \bar{U}_2 & \\ & & \bar{U}_4^\dagger \bar{U}_4 \end{pmatrix} = \mathbb{1}.
$$

This is the decomposing algorithm of shifting the canonical center to the right. If we want to shift leftward, an analgous series of steps follow.

**Left shifting of canonical center:** We will illustrate the reverse process: Given $T^2$ and $M^2$, we can determine $\tilde{T}^2$ and $M^1$ shown from Fig. 13(c) (b) and (a).

The factorization would be represented by



After contraction of $\tilde{T}^2$ and $F^1$, we would reshape into $\tilde{W} \in \mathbb{R}^{d\chi_R \times \chi_L}$ as

$$
\tilde{W} = \begin{pmatrix} A^T & 0 \\ 0 & C^T \\ B^T & 0 \\ 0 & 0 \\ 0 & 0 \\ 0 & 0 \\ 0 & D^T \\ 0 & E^T \end{pmatrix},
$$

which is a reshape of $W$ matrix when merging the site index with the right link index. To simplify our notation, we can permute the rows so that they're grouped based on the subspace structure of the merged index and whole empty rows are removed

$$
\tilde{W} = \begin{pmatrix} \tilde{W}_1 \\ \tilde{W}_2 \end{pmatrix},
$$

with $\tilde{W}_1 = \begin{pmatrix} A^T & 0 \\ B^T & 0 \end{pmatrix}$ and $\tilde{W}_2 = \begin{pmatrix} 0 & C^T \\ 0 & D^T \\ 0 & E^T \end{pmatrix}$. The SVD of them would yield

$$
\text{svd}\left( \begin{pmatrix} A^T \\ B^T \end{pmatrix} \right) \approx \begin{pmatrix} \bar{Y}_A \\ \bar{Y}_B \end{pmatrix} \times \bar{\Lambda}_1 \bar{V}_1^\dagger, \tag{28}
$$

$$
\text{svd}\left( \begin{pmatrix} C^T \\ D^T \\ E^T \end{pmatrix} \right) \approx \begin{pmatrix} \bar{Y}_C \\ \bar{Y}_D \\ \bar{Y}_E \end{pmatrix} \times \bar{\Lambda}_2 \bar{V}_2^\dagger. \tag{29}
$$

The $\bar{Y}_A$ and $\bar{Y}_B$ are upper and the lower part of the corresponding truncated $\bar{U}_1$ after SVD. Similarly, $\bar{Y}_C, \bar{Y}_D, \bar{Y}_E$ are extracted of $\bar{U}_2$ from the SVD. The factorization of $\tilde{W} = \bar{U}\bar{F}^\dagger$ could be written as

$$
\tilde{W} = \begin{pmatrix} A^T & 0 \\ 0 & C^T \\ B^T & 0 \\ 0 & 0 \\ 0 & 0 \\ 0 & 0 \\ 0 & D^T \\ 0 & E^T \end{pmatrix} = \begin{pmatrix} \bar{Y}_A & 0 \\ 0 & \bar{Y}_C \\ \bar{Y}_B & 0 \\ 0 & 0 \\ 0 & 0 \\ 0 & 0 \\ 0 & \bar{Y}_D \\ 0 & \bar{Y}_E \end{pmatrix} \times \begin{pmatrix} \bar{\Lambda}_1 \bar{V}_1^\dagger & 0 \\ 0 & \bar{\Lambda}_2 \bar{V}_2^\dagger \end{pmatrix}.
$$

We can get the new $\tilde{T}^2$ by reshaping $\bar{U}$

$$
\tilde{T}^2 = \begin{pmatrix} \bar{Y}_A^\dagger & 0 & \bar{Y}_B^\dagger & 0 \\ 0 & \bar{Y}_C^\dagger & 0 & 0 \end{pmatrix}_{x_2=0} \begin{pmatrix} 0 & 0 & 0 & 0 \\ 0 & 0 & \bar{Y}_D^\dagger & \bar{Y}_E^\dagger \end{pmatrix}_{x_2=1},
$$

and $F^1$ would be given by

$$F^1 = \begin{pmatrix} \bar{V}_1\bar{\Lambda}_1 & 0 \\ 0 & \bar{V}_2\bar{\Lambda}_2 \end{pmatrix}.$$

We can verifiy that $\tilde{T}^2$ fullfills the canonical condition

$$\sum_i \tilde{T}^{2(i)}(\tilde{T}^{2(i)})^\dagger = \begin{pmatrix} \bar{Y}_A^\dagger\bar{Y}_A + \bar{Y}_B^\dagger\bar{Y}_B & \\ & \bar{Y}_C^\dagger\bar{Y}_C + \bar{Y}_D^\dagger\bar{Y}_D + \bar{Y}_E^\dagger\bar{Y}_E \end{pmatrix} = \begin{pmatrix} \bar{U}_1^\dagger\bar{U}_1 & \\ & \bar{U}_2^\dagger\bar{U}_2 \end{pmatrix} = \mathbb{1}.$$

## 6 Constrained optimization

The ability to factorize and compress tensors via SVD appears in many of the successful optimization algorithms used in many-body physics, and more recently machine learning [15–21]. We will exploit this property of the MPS to employ an optimizer on top of it for the purpose of solving constrained combinatorial optimization problems of the form (1). The approach taken here is inspired by the work [10]. There it was shown how to embed generic equality constraints into a $U(1)$ symmetric tensor network and use that as an initial ansatz to be used in conjunction with a variant of the *generator-enhanced optimization* (GEO) framework of Ref. [22]. While alternative optimizers like the density matrix renormalization group (DMRG) and imaginary time evolution could also be employed, they necessitate conversion of the optimization function into QUBO format and generally apply only to local functions. The approach used here circumvents these limitations, while preserving the flux of each tensor - i.e. the optimization occurs within the feasible solution space. Our optimization strategy mirrors that of Algorithm 1 in [10], targeting arbitrary cost functions using a Born machine (BM) representation $p(\boldsymbol{x}) = |\psi(\boldsymbol{x})|^2/Z$, where $\psi$ is the MPS and $Z$ normalizes the probability over all binary vector states in $\{0,1\}^N$. At every iteration the goal is to minimize the loss function

$$\mathcal{L} = -\sum_{\boldsymbol{x} \in \mathcal{T}} \log p(\boldsymbol{x}), \tag{30}$$

where $\mathcal{T}$ comprises samples drawn from a Boltzmann distribution reflecting historical cost data, $p_T = e^{-C(\boldsymbol{x})/T} / \sum_{\boldsymbol{x} \in \mathcal{T}_{\text{tot}}} e^{-C(\boldsymbol{x})/T}$, with $\mathcal{T}_{\text{tot}}$ the set of samples extracted from the model distribution at all past iterations. (Note that for cost functions that are expensive to evaluate, one may call the cost function once per sample and store the result in memory for later access when updating the Boltzmann distribution). The process begins by constructing a constrained MPS using Algorithm 2, followed by iterative training to refine this model via (30). We only use a single MPS gradient step per iteration as in [10]; thus our goal is not to minimize (30) to convergence. Crucially, if the minimum cost of newly generated samples does not improve upon the previous iteration, the MPS is reset to its initial state to prevent overfitting and improve diversity of samples. Additionally, performance is enhanced by implementing a temperature annealing schedule over iterations within the Boltzmann distribution. In our case we choose the following simple schedule: $T_{\text{ini}}/t$, $t = 1, 2, \cdots, t_{\text{max}}$, with the initial temperature $T_{\text{ini}}$ set according to the standard deviation of initial costs. We note that a similar approach in spirit was proposed in the *variational neural annealing* framework of Ref. [23]. However, a key distinction between our method and variational neural annealing (as well as GEO) is that our goal at each iteration is not to learn the target Boltzmann distribution over the entire space. Instead, we use Eq. (30) with the sole goal of producing slightly better samples (of lower cost) than those in the training set. In practice, this involves training with just a single gradient descent step on the MPS, as opposed to the multiple steps required for modeling the Boltzmann distribution (in general). Essentially, the success of our algorithm hinges on the use of many iterations.

**Training.** To minimize Eq. (30) we use the training algorithm from Ref. [16], with a one site gradient update. This algorithm preserves block sparsity [10], so it can be used in the presence of constrained tensors in conjunction with the new canonical form from this work. Each gradient descent step on the MPS is comprised of a full forward and backward sweep applying one site gradient update on each tensor. Each such gradient update is followed by a compression step to keep the resulting tensor low-rank. This is achieved by performing an SVD decomposition on joint blocks as detailed in the previous section, and dropping the singular values below a cutoff $\epsilon$. This can have the dramatic effect of removing QRegions that do not contribute significantly in the factorization process.

**Sampling.** One benefit of the optimization procedure used here is that it exploits the fast and *perfect* sampling property of MPS [24]. Such sampling procedure preserves block sparsity as well.

**Select.** At every iteration one must sample from the dictionary of all collected samples according to the Boltzmann distribution. The number of samples is fixed throughout the iterations. Whenever we find that the minimum cost of collected samples is not lower than that of the previous iteration, $C_{\min}(t) \geq C_{\min}(t-1)$, we reset the MPS to its initial value. This is followed by sampling from it and replacing a small fraction of the training samples from the current iteration by samples from this new MPS.

---

**Algorithm 3** Tensor network optimizer for linear constraints

---

**Input**  Callback cost function $y = C(\boldsymbol{x})$, $\boldsymbol{x} \in \{0,1\}^{\otimes N}$, linear constraints $\{\mathbf{A}, \boldsymbol{\ell}, \boldsymbol{u}\}$, number iterations $t_{\max}$, SVD cutoff $\epsilon$
**Output**  $\boldsymbol{x}^* \approx \mathrm{argmin}(C(\boldsymbol{x}))$

1: $\psi_0 \leftarrow \textsc{ConstraintsToMPS}(\mathbf{A}, \boldsymbol{\ell}, \boldsymbol{u})$   (▷) Algorithm 2
2: $t \leftarrow 0$
3: $\psi \leftarrow \psi_0$
4: $\mathcal{T} \leftarrow \textsc{Sample}(|\psi|^2)$   (▷) Extract training set via perfect sampling from constrained MPS BM
5: $\mathcal{T}_{\mathrm{tot}} \leftarrow \mathcal{T}$   (▷) Create a set of unique samples seen so far from training set
6: $C_{\min}(0) \leftarrow \infty$
7: $C_{\min}(1) \leftarrow \mathbf{min}(\{C(\boldsymbol{x}): \boldsymbol{x} \in \mathcal{T}\})$
8: $C_{\mathrm{cum, min}} \leftarrow C_{\min}(1)$   (▷) Minimum cost observed so far
9: **for** $1 \leq t \leq t_{\max}$ **do**
10:    $D[\boldsymbol{x} \in \mathcal{T}_{\mathrm{tot}}] \leftarrow e^{-C(\boldsymbol{x})/T_t} / \sum_{\boldsymbol{x} \in \mathcal{T}_{\mathrm{tot}}} e^{-C(\boldsymbol{x})/T_t}$   (▷) Append samples and their probabilities to the dictionary
11:    $\mathcal{T} \leftarrow \mathrm{sample}(D[\boldsymbol{x}])$   (▷) Sample from dictionary and construct training set
12:    **if** $C_{\min}(t) < C_{\min}(t-1)$ **then**
13:       **if** $C_{\min}(t) < C_{\mathrm{cum,min}}$ **then**
14:          $C_{\mathrm{cum,min}} \leftarrow C_{\min}(t)$
15:       **end if**
16:    **else**
17:       $\psi \leftarrow \psi_0$   (▷) Rebuild constrained MPS
18:       $\mathcal{T} \leftarrow \textsc{Select}(\mathcal{T}, \textsc{Sample}(|\psi|^2))$   (▷) Construct new training set from previous set and $|\psi_0|^2$ samples
19:    **end if**
20:    $\psi \leftarrow \textsc{Train}(\psi, \mathcal{T}, \epsilon)$
21:    $\mathcal{T} \leftarrow \textsc{Sample}(|\psi|^2)$
22:    $C_{\min}(t+1) \leftarrow \mathbf{min}(\{C(\boldsymbol{x}): \boldsymbol{x} \in \mathcal{T}\})$
23:    $\mathcal{T}_{\mathrm{tot}} \leftarrow \mathcal{T}$   (▷) Append unique and unseen samples to $\mathcal{T}_{\mathrm{tot}}$
24: **end for**

**Return**  $\mathrm{argmin}(C(\boldsymbol{x}): \boldsymbol{x} \in \mathcal{T}_{\mathrm{tot}})$, $C_{\mathrm{cum,min}}$.

---

# 7   Results

To test the performance of our algorithm, we consider the Quadratic Knapsack Problem (QKP), a well-known combinatorial optimization problem that serves as a meaningful benchmark for constrained optimization methods. Unlike its linear counterpart, which can often be solved efficiently using dynamic programming in pseudo-polynomial time, QKP remains strongly NP-hard even for small integer weights [25], making it a computationally challenging test case. Additionally, QKP naturally fits our framework, as our method distinguishes between constraint enforcement and black-box objective function optimization—allowing us to handle nonlinear objective functions like QKP without modifying our encoding of constraints.

Beyond theoretical complexity, QKP is widely studied in operations research and resource allocation problems, making it a practical benchmark. In its most general form can be formulated as minimizing a quadratic objective function subject to an inequality constraint of the following form:

$$
\begin{aligned}
\min\ &\boldsymbol{x} \cdot \mathbf{Q}\boldsymbol{x}\,, \\
&\boldsymbol{w} \cdot \boldsymbol{x} \le W\,,
\end{aligned}
\tag{31}
$$

where $\boldsymbol{x} \in \{0,1\}^N$ is a binary vector of length $N$, $\mathbf{Q}$ is a $N \times N$ matrix of integers, and $\boldsymbol{w}$ is a vector of nonnegative integers of length $N$, and $W$ is the knapsack capacity. For our benchmark, we employ the open-source SCIP solver, a premier tool for integer nonlinear programming that operates within the *branch-cut-and-price* framework [26]. This algorithm provides global solution optimality guarantees through primal (lower bound) and dual (upper bound) solutions, where the primal solution represents the most favorable solution identified thus far. Here we will use the SCIP solver through the JuMP.jl Julia package [27]. Our tensor network simulations use a forked ITensor.jl [28] version as a backend. It can be found as a submodule in our project repository [11]. All numerical simulations were carried out on an Apple M2 Pro chip.

We examine a set of problems as specified by equation (31), in which the elements of $\mathbf{Q}$ and $\boldsymbol{w}$ are randomly selected as integers from uniform distributions within the intervals $[-5,5]$ for $\mathbf{Q}$, and $[0,5]$ for $\boldsymbol{w}$. The knapsack capacity is chosen as $W = \lfloor N/4 \rfloor$. We will carry out 10 such experiments for various problem sizes, $N = \{50, 100, 200, 400\}$. The experiment setup consists of running Algorithm 3 for 75 iterations. For a given problem size, each of the 10 different runs might take different wall-clock times. Thus we average the time taken across those runs and use that time as the allotted time for the SCIP solver to solve each instance. The times used correspond to 230 secs. for $N = 50$, 500 secs. for $N = 100$, 1100 secs. for $N = 200$, and 2500 secs. for $N = 400$. Note that to make the comparison as direct as possible we used a single thread in both algorithms. The TN solver could benefit not only from multiple threads (used e.g. when computing $l_i$ and $\tilde{l}_i$ on independent threads, as well as for handling multiple data batches during training), but also the use of GPUs. The charge complexity of the knapsack constraint is low (e.g. for $N = 400$, $Q \approx 100$). This results in relatively fast optimization times. Further, for all our experiments we choose the same optimization parameters: cutoff $\epsilon = 1E{-}4$, learning rate 0.05, and an initial temperature $T_1 = 2.5N$, corresponding roughly to the std of a set of feasible bitstrings randomly chosen. We also choose $|\mathcal{T}| = 400$ corresponding to both number of training and output samples from the MPS. When $C_{\min}(t) \ge C_{\min}(t-1)$, we replace 40 samples in the current training set with 40 samples from $\psi_0$. See [11] for details on the implementation.

The performance analysis between our TN solver and SCIP on the quadratic knapsack problem, as illustrated in Figure 14, demonstrates the comparative efficiency and effectiveness of the TN solver across various problem sizes. The left panel of the figure presents a box plot depicting the percentage improvement of the TN solver over the SCIP solver. The percentage improvement is calculated as (TN best - SCIP best)/ SCIP best $\times 100\%$, where *best* corresponds

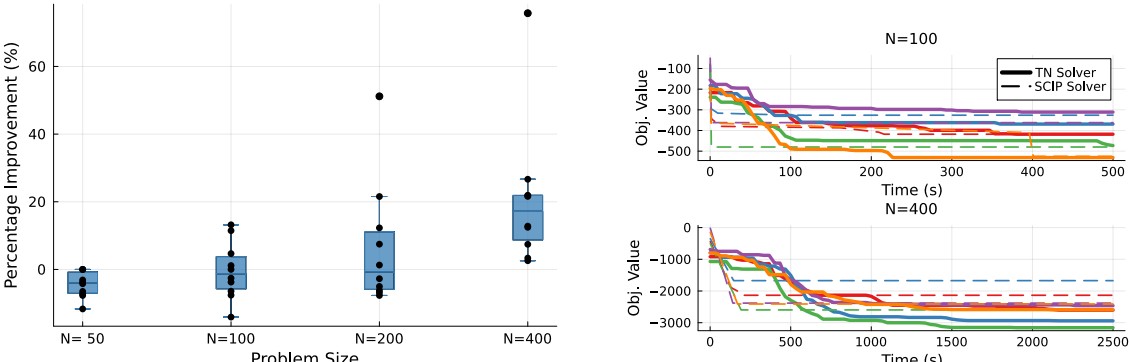

Figure 14: **Performance comparison of our TN solver *vs.* SCIP on quadratic knapsack.** (*Left*): Percentage improvement (TN best - SCIP best)/ SCIP best ×100% as a box plot along with the results for each of the ten instances in dots, showcasing that the TN solver consistently outperforms SCIP in larger problem instances ($N \geq 200$). (*Right*): Solver's best objective value over time on five random problem realizations for problem sizes $N = 100$, $N = 400$.

to the best objective value found by the solver. This metric highlights the relative performance, showing that the TN solver consistently achieves better results, especially in larger problem instances where $N \geq 200$. We note that for $N = 50$, the SCIP solver finds the exact global optimal solution for all instances within the allotted time, so the performance improvement is capped at 0. Nevertheless, TN solver is able to find the global optimum in three out of the ten instances.

The right panel complements these findings by depicting the best objective value found by both solvers over time across five randomly selected problem realizations. This time-based view provides insights into the operational behavior of the solvers, showing that the TN solver not only reaches lower cost solutions at larger problem sizes but also does so in a more stable and predictable manner. In particular, we've found that while the TN solver has a slow start at finding *good* candidate solutions, it quickly surpasses SCIP for all instances at $N = 400$, and half of the instances at $N = 100$. Moreover, SCIP would tend to get stuck and only find very few candidate solutions, while the TN solver would always produce feasible solutions at all times, and quite a few below the best solution found by SCIP. Hence not only does the TN solver find better solutions for larger problems, it is also able to find a greater variety of such good samples.

# 8 Conclusions and outlook

In this work we have introduced a novel way of embedding arbitrary discrete linear constraints into a tensor network using ideas from constraint programming and inspired by the theory of $U(1)$ symmetric tensor networks. The main intuition behind our method is that global discrete linear constraints can be decomposed into local ones by the introduction of new local degrees of freedom, that we term here *quantum regions*. These follow a certain fusion rule, enabling a structured representation of constraints. While prior works have explored connections between tensor networks and linear constraints (e.g., [29–34]), our approach is significantly more general. Unlike methods that focus solely on local constraints (e.g., CNF formulas in SAT problems with at most three variables per clause [29, 30]) or equality constraints [10], our framework applies to a broader class of constraints. Furthermore, in contrast to approaches

requiring manual constraint-specific encoding (e.g., [33]), our method provides a systematic and automated way to embed constraints into tensor networks. Further, by representing the MPS links in terms of quantum regions, we gain computational savings, and we suspect that no further compression can be achieved without losing part of the feasible space. Lastly, we have introduced a new canonical form for constrained MPS and used it as part of an optimization algorithm, Algorithm 3, which is inspired by the GEO framework of Ref. [22], and the constrained embedding and rebuild steps of Algorithm 1 of Ref. [10]. The present approach is not limited to equality constraints, and the rebuild step is chosen judiciously based on the current samples' costs. Crucially, our optimization cycles include annealing, emphasizing the *quantity* of these cycles over the *quality* of individual optimizations of the loss function, Eq. (30). For instance, just one training descent step per cycle suffices to achieve great results after many cycles.

Future work is directed both at applying some of these techniques to new domains, as well as on improving some of the steps developed here. The constraints discussed in this work are not just restricted to combinatorial optimization problems. One potential application of our constrained tensor network is precisely in the context of quantum many-body physics where the $U(1)$ global symmetry may be mildly broken, such as in open systems with constrained particle conservation. A classic example of $U(1)$ conservation is magnetization in spin chains: $\sum_i \hat{S}_i^z = M$, where $\hat{S}_i^z$ corresponds to the local magnetization of spin $i$ along the $z$ direction, with eigenvalues $s_i^z \in \{-1/2, 1/2\}$. Such conservation laws, while physically motivated, are often an idealization. Various sources of noise in an experimental system can break, even if only mildly, such conservation laws. A relevant question in this context is under which setups one would have $M_\ell \leq \sum_i \hat{S}_i^z \leq M_u$, where $M_{\ell/u}$ are some fixed lower/upper bounds to the total magnetization. Such constraints could be easily handled by our tensor network approach.

Further improvements of this work involve better handling of multiple constraints. The scaling for this seems to be exponential in the number of constraints, by means of analyzing the facility location problem. Finding ways to extract QRegion blocks from data, similar to the way one can extract quantum numbers from data as proposed in Ref. [10], would certainly be a way to ameliorate the bottlenecks from embedding all the constraints exactly.

## Acknowledgments

We thank Alejandro Perdomo-Ortiz for fruitful collaboration on past projects. We also thank Lei Wang and Alejandro Pozas-Kerstjens for feedback and helpful comments on our work.

## A   Proof of optimal factorization

In this section, we will solve the optimal factorization in Eq. 12.

$$\min_{\bar{U}_i, \bar{F}} \left( \left\| W - \bar{U}\bar{F} \right\|^2 \right), \quad \text{s.t.} \quad \bar{U}_i^\dagger \bar{U}_i = I_i, \quad \forall i. \tag{A.1}$$

Due to the isometry constraint, we add Lagrange multipliers $\bar{\Lambda}_i$ into the target function $f\left(\bar{U}_i, \bar{F}\right)$

$$f\left(\bar{U}_i, \bar{F}\right) = \mathrm{Tr}\left((W - \bar{U}\bar{F})(W - \bar{U}\bar{F})^\dagger\right) + \mathrm{Tr}\left(\bar{\Lambda}_i(\bar{U}_i^\dagger \bar{U}_i - I_i)\right). \tag{A.2}$$

In general, the Lagrange multipliers should be symmetric matrix. Since any unitary rotations in the subspace spanned by columns of $\bar{U}_i$ are equivalent. We choose the rotation so that the

multiplers are diagonalized and represent by $\bar{\Lambda}_i$.

$$
\begin{aligned}
f\left(\bar{U}_i, \bar{F}\right) &= \mathrm{Tr}\left(WW^\dagger - \bar{U}\bar{F}W^\dagger - W\bar{F}^\dagger\bar{U}^\dagger + \bar{U}\bar{F}\bar{F}^\dagger\bar{U}^\dagger\right) + \mathrm{Tr}\left(\bar{\Lambda}_i(\bar{U}_i^\dagger\bar{U}_i - I_i)\right) \\
&= \mathrm{Tr}\left(WW^\dagger - \bar{U}\bar{F}W^\dagger - W\bar{F}^\dagger\bar{U}^\dagger + \bar{F}\bar{F}^\dagger\bar{U}^\dagger\bar{U}\right) + \mathrm{Tr}\left(\bar{\Lambda}_i(\bar{U}_i^\dagger\bar{U}_i - I_i)\right) \qquad (A.3) \\
&= \mathrm{Tr}\left(WW^\dagger - \bar{U}\bar{F}W^\dagger - W\bar{F}^\dagger\bar{U}^\dagger + \bar{F}\bar{F}^\dagger\right) + \mathrm{Tr}\left(\bar{\Lambda}_i(\bar{U}_i^\dagger\bar{U}_i - I_i)\right).
\end{aligned}
$$

The second equal sign holds due to cyclic identity in the trace. The third equal sign holds due to the isometry constrain of $\bar{U}$. We can then split $\bar{U} = \oplus_i \bar{U}_i$ and insert identity $\bar{U} = \sum_i P_i \bar{U} P_i = \oplus_i \bar{U}_i$ and $\sum_i P_i = I$ into the loss.

$$
\begin{aligned}
f\left(\bar{U}_i, \bar{F}\right) &= \mathrm{Tr}\left(WW^\dagger - \bar{U}\bar{F}W^\dagger - W\bar{F}^\dagger\bar{U}^\dagger + \bar{F}\bar{F}^\dagger\right) + \mathrm{Tr}\left(\bar{\Lambda}_i(\bar{U}_i^\dagger\bar{U}_i - I_i)\right) \qquad (A.4) \\
&= \mathrm{Tr}\left(WW^\dagger - \sum_i P_i\bar{U}P_i\bar{F}W^\dagger - W\bar{F}^\dagger\sum_i P_i\bar{U}^\dagger P_i + \sum_i P_i\bar{F}\bar{F}^\dagger P_i\right) + \mathrm{Tr}\left(\bar{\Lambda}_i(\bar{U}_i^\dagger\bar{U}_i - I_i)\right),
\end{aligned}
$$

where $P_i$ is the projector into subspace defined by row index block structure.

$$
f\left(\bar{U}_i, \bar{F}\right) = \mathrm{Tr}\left(WW^\dagger - \sum_i \bar{U}_i\bar{F}_i W_i^\dagger - \sum_i W_i\bar{F}_i^\dagger\bar{U}_i^\dagger + \sum_i \bar{F}_i\bar{F}_i^\dagger\right) + \mathrm{Tr}\left(\bar{\Lambda}_i(\bar{U}_i^\dagger\bar{U}_i - I_i)\right), \quad (A.5)
$$

where $W_i = P_i W$ are the rows of $W$, and $F_i = P_i F$, are projected on $i$-th subspace. To minimize $f\left(\bar{U}_i, \bar{F}\right)$, we require

$$
\frac{\partial f}{\partial \bar{U}_i^\dagger} = -W_i\bar{F}^\dagger + \bar{U}_i\bar{\Lambda}_i = 0, \qquad (A.6)
$$

$$
\frac{\partial f}{\partial \bar{F}_i^\dagger} = -\bar{U}_i^\dagger W_i + \bar{F}_i = 0. \qquad (A.7)
$$

From Eq. A.7, we can get

$$
\bar{F}_i = \bar{U}_i^\dagger W_i, \qquad (A.8)
$$

when substitution into Eq. A.6 we get

$$
W_i W_i^\dagger \bar{U}_i = \bar{U}_i\bar{\Lambda}_i. \qquad (A.9)
$$

This is the eigenvalue decomposition of $W_i W_i^\dagger$, which is equivalent to SVD decomposition of $W_i$. Suppose

$$
W_i = U_i \Lambda_i V_i^\dagger.
$$

The symbols without bar means no truncation has been applied.

$$
\begin{aligned}
\bar{F}_i &= \bar{U}_i^\dagger W_i \\
&= \bar{U}_i U_i \Lambda_i V_i^\dagger \qquad (A.10) \\
&= \bar{\Lambda}_i \bar{V}_i^\dagger.
\end{aligned}
$$

The loss would be given by

$$
\sum_i (\Lambda_i - \bar{\Lambda}_i)^2.
$$

So the optimal truncation is selecting the largest $\chi$ values among all $\lambda_i^j$ values.

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
