# Peer review of "Cons-training Tensor Networks: Embedding and Optimization Over Discrete Linear Constraints"

_SciPost Physics, doi:SciPost Phys. 18, 192 (2025)_

## Round 3 · Referee Report · Anonymous (Referee 1) · 2025-3-7

Strengths

  1. The QRegions approach is clearly explained.
  2. Numerical benchmarks support the claims.

Weaknesses

  1. It may be hard for non-specialists to understand precisely what challenge this work aims to tackle. This is not made sufficiently clear in the introduction.
  2. Some statements in the text are sweeping or not sufficiently rigorous - see requested changes below.

Report

This manuscript describes an MPS representation adapted to the solution of combinatorial optimization problems with hard equality or inequality constraints. The main technical contribution is a methodology based on QRegions that allows one to encode the constraints of an optimization problem while at the same time eliminating parts of the Hilbert space that are guaranteed to be empty of feasible solutions. Importantly, the approach allows for canonical form MPSs that are explicitly Schmidt-decomposed and hence enjoy optimal compression via standard MPS techniques. The authors then use a previously introduced heuristic that optimizes these MPSs iteratively, such that their sampling produces progressively better solutions.

The description of the methods is clear and the examples and figures help the reader to follow the steps. The methods developed are then benchmarked on instances of the quadratic knapsack problem. Overall, this work is a valuable addition to the arsenal of MPS / TN techniques for computational problems.

One troubling aspect of this work is that its claims are at times vague and potentially hide inherent limitations of the methods introduced. I detail my reservations in this respect in the requested changes below. I ask that the authors address these concerns before I can recommend publication.

Requested changes

Main remarks: A. In the beginning of page 9, the authors write: "Hence, by working with QRegions we guarantee the most compact representation of a constrained MPS. We remark that we are including QNs into each QRegion that are never part of the solution space." First, how can the authors guarantee that this representation is the most compact one? Later in the same paragraph, the authors claim that the QRegions can be chosen according to computational efficiency, but this suggests that the representation chosen may be the most practical, not the most compact. Second, if QRegions include QNs that are not in the solution space, how are spurious non-solutions avoided in the sampling process? In principle, allowing non-solutions in multiple QRegions will lead to a multiplicative overhead of non-solutions in the MPS overall, which could severely slow down the search for solutions in the optimization step.

B. The authors do not report the complexity of Algorithm 2. The setting of Eq. (1) allows for arbitrarily complex problems in classes NP / #P (e.g., 3SAT). For example, enforcing constraints corresponding to a #P problem with exponentially many solutions would yield a volume-law MPS - see, e.g., 1302.2826, Figs. 2 and 3. This contradicts the complexity statements in Sec. IV, if the latter are not better qualified. Also, this contradicts the statement of the authors in the first paragraph of Sec. VI, where they claim that their approach circumvents the limitations of other optimization methods: in both cases, non-local constraints will lead to volume-law states.

C. The discussion of charge complexity is also problematic. The authors claim that it grows generically polynomially with the number of variables. This clearly cannot be the case in general, unless the authors are also implying that P=NP. Therefore, if the constraints enforced on the MPS correspond to hard problems in NP, then the charge complexity (and the complexity of contracting the MPS) must be exponential in the worst case. On the other hand, what the authors probably have in mind is to constrain the MPS into a feasible subspace and not the solution space directly. If that's the case, then the difference between feasible vs solution subspace should be made (more) explicit in the introduction.

D. In Sec. VII, the authors choose to study instances of the quadratic knapsack problem. The choice of this problem (and the specific choice of instances of the problem) is not justified. Why is this problem interesting? Why is it hard? Knapsack instances are easy on average. What would be the performance of the algorithm on more common benchmarks, such as random 3SAT instances, for example? (Note that a 3SAT constraint without negations is straightforwardly written as the inequality x1+x2+x3>0.)

Points to check: 1. Fig. 3(b): the horizontal axis is labeled l3. Should it be l1? 2. Eq. (3) and many other equations: improper punctuation before, in, and after. 3. Sec. IIIB, first paragraph: l_i used to define both lower bounds and link indices. 4. Fig. 8: labeling the axes would be helpful. 5. Fig. 9(e), bottom right: horizontal axis is labeled l2. Should it be l1? 6. Fig. 12: this figure is hard to read. Symbols too small and indistinguishable. 7. 4th equation after Eq. (27): Should it be U4 instead of U3?

Recommendation

Ask for minor revision

  • validity: high
  • significance: good
  • originality: good
  • clarity: high
  • formatting: good
  • grammar: good

Author:  Javier Lopez Piqueres  on 2025-03-26  [id 5318]

(in reply to Report 1 on 2025-03-07)

Warnings issued while processing user-supplied markup:

  • Inconsistency: plain/Markdown and reStructuredText syntaxes are mixed. Markdown will be used.
    Add "#coerce:reST" or "#coerce:plain" as the first line of your text to force reStructuredText or no markup.
    You may also contact the helpdesk if the formatting is incorrect and you are unable to edit your text.

We thank the referee for the thoughtful comments, which helped us improve the clarity of our work which will be shared in the resubmission stage. Below we reply to each of the remarks:

A. Regarding the claim of compactness: We have revised the manuscript to clarify that by “most compact representation,” we refer specifically to compactness in terms of bond dimension, which is the standard measure of representational complexity in tensor networks. Our approach does not necessarily minimize memory usage or parameter count in every part of the MPS, but it ensures that no increase in bond dimension is incurred by the use of QRegions. This point is now made explicit in the future version’s revised paragraph (page 9):

“Hence, by working with QRegions, we guarantee the most compact representation of a constrained MPS in terms of bond dimension…”

When deriving the fusing rules and QRegions, it is always possible to break each QRegion into smaller pieces. In extreme cases, each QRegion contains only one QN. In this extreme case, we can also guarantee that all constraints are encoded; however, this is not the optimal choice. For example, in Fig.~8(b), if we represent all QNs, we need more than 17 QRegions. Even though some of these QNs will be filled in after the forward sweep in Fig.~9(c), we still need to keep track of 17 block sparse structures, and the minimum bond dimension is at least 1 for each block, leading to $D \geq 17$. This introduces a significant overhead when $M$ and $N$ are large. Therefore, we always seek to find the minimal number of QRegions and conditions necessary to represent each link. To ensure that the constraints for $l_3$ are always maintained, we must at least keep track whether it belongs to the green, yellow, or blue region. If all flux is assigned to the last tensor, the minimum bond dimension would be $D = 3$. However, if we move the canonical center to other sites (Sec.~V), this can be reduced to $D = 2$. We cannot further combine any two regions into one, since different QRegions exhibit different fusion behaviors conditioned on input $x_4$.

Regarding the inclusion of invalid QNs in QRegions and sampling correctness: We fully agree that including QNs not part of any valid solution path could, in principle, raise concerns about efficiency. To clarify, these non-solution QNs do not indicate constraint violations. We refer to them as non-solution (or virtual in the text) QNs not because they violate constraints, but because they cannot be realized by any bit string—even without constraints. For example, in Fig.~8(a), for $l_4$, all 12 QNs satisfy all constraints; however, only 4 of them can be realized in Fig.~9(d). This means that there is no valid solution corresponding to certain QNs.

When deriving the QRegions and fusion rules, we aim to explore all possibilities. The sweep process will validate these $QNs$, ruling out some of them during the sweep. Similarly, in Fig.~8(b), when deriving the QRegion for $l_3$, we consider all black-dotted QNs, but only 4 QNs survive the sweep process (see Fig.~9(c)).

We now explicitly state in the forthcoming version that the fusion rules governing the contraction of constrained MPS tensors ensure that only feasible global configurations contribute to the overall wavefunction. Even if a QRegion includes QNs that are never visited by any valid solution, the fusion constraints ensure that any path passing through such states will have zero amplitude.

This point is now clarified in the manuscript with a concrete example (page 9):

“Crucially, although QRegions may include non-valid QNs, the fusion rules governing the tensor contractions guarantee that only feasible global configurations contribute to the MPS…”

We hope this clarification resolves the concern regarding the potential overhead from invalid QNs. We also added a brief note highlighting that a formal comparison between different QRegion encodings (e.g., unions of valid QNs versus geometric regions) in terms of practical performance remains an interesting avenue for future work.

B. We thank the referee for mentioning the connection between volume-law MPS and #P problems with exponentially many solutions, as highlighted by the referenced work. Indeed, we acknowledge that such volume-law scaling can arise for certain classes of constraints. However, in our experience, many typical constraints that occur in constrained combinatorial optimization, including global constraints, can be captured with only polynomial resources—both for constructing the MPS and for contracting it to count solutions.

For example, consider the (global) cardinality constraint of equality type

\sum_{i=1}^N x_i = k,

which has on the order of \frac{1}{\sqrt{N}} exp(N) solutions at k = N/2. Despite this exponential number of solutions, the maximum bond dimension remains linear in N, specifically \chi = N/2 + 1. This is shown in Eq. (5) of [2211.09121] and illustrated in Fig. 11 of our work (where we also extend the idea to inequalities). Consequently, the total complexity of counting these solutions scales as O(N^2) (see the paragraph after Eq. (7) in our manuscript, with Q = N/2). In contrast, Figs. 2–3 of [1302.2826] analyze how the bond dimension grows with the number of constraints (clauses). As expected, the resulting scaling can become exponential, which we also observe in Fig. 12 (bottom left) of our work. However, we find empirically that for certain constraints—such as those with a fixed number of facility-location-type constraints—the complexity grows only polynomially with number of variables, as shown in Fig. 12 (top left).

Note also that [1302.2826] specifically addresses 2SAT, in which each constraint involves exactly two variables. By contrast, our problem setting allows for global constraints involving an extensive number of variables, and we do not restrict ourselves to CNF form. To the best of our knowledge, all prior tensor-network approaches for SAT (some of which we cite in Sec. VIII) handle only a few variables per clause in CNF form, since each clause contributes a tensor whose rank grows with the clause size, and this has an exponential overhead. In contrast, our framework decomposes an arbitrary global constraint into local ones in terms of order 3 tensors (MPS tensors), thereby enabling the efficient treatment of many structured, global constraints that are not limited to CNF.

Finally, regarding the complexity of Algorithm 2: it essentially follows from Algorithm 1. Once Algorithm 1 identifies all the relevant QRegions (and thus determines the bond dimension), Algorithm 2 simply labels the corresponding tensor blocks (setting them, for instance, to 1). Hence, the overall cost of Algorithm 2 is dominated by lines 2 and 3. The scaling of Algorithm 1 is discussed in Section IVA.

We hope this clarifies why certain #P-hard constraints can indeed induce volume-law MPS scaling, whereas other, more common or structured constraints remain tractable. We have updated our discussion in Sections IV and VIII to reflect these distinctions more explicitly.

C. We thank the referee for noting that our claims of polynomial scaling would be misleading if taken to apply universally to all NP-hard constraints. We fully agree that in the worst case, when constraints encode NP or #P problems, the charge complexity (and hence the MPS bond dimension) can grow exponentially with the number of variables, in line with known complexity results. We do not intend to imply that P=NP.

Instead, our empirical observations of polynomial (often linear) growth in charge complexity are tied to structured constraints commonly found in practical combinatorial optimization problems. For instance, the global cardinality constraint discussed above \sum_i x_i = k — which has an exponentially large solution set — nonetheless leads to a linear bond dimension (\chi = k+1), as demonstrated in the analytic example we provided (and as discussed in [2211.09121]). Many other constraints (including mild modifications of the cardinality constraint, such as certain knapsack-type constraints) exhibit similar behavior in practice, showing polynomial bond dimension and polynomial-time contraction. We view the cardinality-constraint case as an indicative example of what one might expect for other structured, global constraints.

We also emphasize that our construction is exact and represents all feasible solutions — not just a subset. Every sample that satisfies the constraints is retained by construction, and no additional (infeasible) configurations appear in the MPS. Hence, we are encoding the solution space (feasibility region), rather than merely a feasible subspace. We have clarified in the revised version that while the worst-case scaling can be exponential for certain pathological constraints, our method is particularly suited to many practical or structured constraints, for which we empirically and analytically observe polynomial scaling in bond dimension.

If one were to construct a family of single constraints that specifically encode NP-hard instances without exploitable structure, we fully agree that this could yield an exponentially growing bond dimension. Indeed, exploring such worst-case families vs. typical or structured families is an interesting direction for future work.

D. We thank the referee for asking about our choice of the Quadratic Knapsack Problem (QKP) as the main test case. There are a few key reasons we selected QKP over other options like 3SAT or simpler knapsack variants:

  1. Quadratic Knapsack is Strongly NP-Hard – While linear knapsack with small integer weights and capacities can often be solved in pseudo-polynomial time by dynamic programming, making it “easy” in many practical cases, the quadratic version remains strongly NP-hard. This captures the inherent difficulty we want to test. – For the specific setting of weights in {0,1,\dots,5} and capacity W = N/4, linear knapsack instances can indeed be solved efficiently by branch-and-bound or dynamic programming, so it would not highlight the advantages of our method. By contrast, non-linear (quadratic) cost functions are far less tractable, even with small integral weights, which makes QKP an interesting benchmark.

  2. Our Method Distinguishes Between Constraints and Cost – We focus on problems with linear constraints (which our approach enforces exactly) but allow an arbitrary objective function. That means we can handle difficult objective functions, including quadratic ones, without changing the way constraints are encoded. – In contrast, 3SAT does not separate out a “cost function” from feasibility constraints—it’s purely a feasibility problem. Our aim is to solve optimization problems (maximize or minimize a cost subject to constraints). Hence, QKP naturally fits our framework.

  3. Practical Relevance and Comparison with a State-of-the-Art Solver – QKP has practical significance (e.g., certain resource allocation or capital budgeting applications). Although specialized solvers exist, they are not nearly as straightforward as linear MIP solvers. – We chose SCIP because it is one of the state-of-the-art open-source solvers for nonlinear integer programs, as documented in [see https://link.springer.com/article/10.1007/s11081-018-9411-8 page 424)]. This allows a fair performance comparison: we encode the same problem in SCIP and in our MPS-based approach.

We hope this clarifies our motivation for focusing on QKP. We have added a brief explanation in the upcoming revised version (Section VII) to better justify this choice. If the referee has further suggestions or specific benchmarks to consider, we would be happy to discuss them.

Lastly, we have amended points 1–7 as suggested and sincerely appreciate the referee’s careful review.

---

## Round 3 · Referee Report · Anonymous (Referee 2) · 2025-4-15

Strengths

An outstanding aspect of this paper is that it provides a substantial amount of detail (including the simulation implementation code), clearly demonstrating the logical flow and ensuring readability.

Weaknesses

Specialized knowledge is required, and the algorithm might be somewhat advanced for nonexpert readers to fully understand.

Report

This is a very interesting paper that introduces the concept of embedding arbitrary discrete linear constraints into a tensor network, borrowing ideas from the finite state machine and inspired by the theory of U(1) symmetry in tensor networks. In this case, the method developed in this paper can be applied to minimizing a quadratic objective function subject to not only linear equations but also inequality constraints of a general form. Considering the success of U(1) symmetry in efficiently handling global constraints, this idea has been widely applied in quantum physics for solving many-body problems—such as computing doped holes in lattice electron models. However, as far as I know, it has rarely impacted areas outside of physics. Applications remain closely tied to the method, but it could potentially have a significant impact on industries dealing with combinatorial optimization problems, such as finance, logistics, and telecommunications. The authors have developed a series of papers introducing tensor network methods into constrained combinatorial optimization problems. This paper represents related progress following their previous work (Ref [10]), where linear equation constraints were treated, demonstrating a continuous and rigorous style of writing. I enjoyed reading this paper and understanding how the authors developed this methodology. Therefore, I suggest this paper for publication.

Requested changes

I have reviewed the comments and responses from the previous round of peer review, and I believe the authors have thoroughly addressed those concerns. The clarity of the current manuscript has significantly improved, and I consider it suitable for publication without further revision.

Recommendation

Publish (surpasses expectations and criteria for this Journal; among top 10%)

  • validity: top
  • significance: high
  • originality: top
  • clarity: high
  • formatting: excellent
  • grammar: excellent

Author:  Javier Lopez Piqueres  on 2025-04-27  [id 5419]

(in reply to Report 2 on 2025-04-15)

We sincerely thank the referee for their thoughtful and positive evaluation of our work. We are very pleased to hear that the referee found the paper interesting, well-written, and clearly presented, and that they recognized its potential impact beyond physics. We are grateful for their recommendation to publish the paper without further revision.

---

## Editorial Decision

published